# Accurate prediction of kinase-substrate networks using knowledge graphs

Vít Nováček[1,7†]*, Gavin McGauran[3], David Matallanas[3], Adrián Vallejo Blanco[3,4], Piero Conca[2], Emir Muñoz[1,2], Luca Costabello[2], Kamalesh Kanakaraj[1], Zeeshan Nawaz[1], Brian Walsh[1], Sameh K. Mohamed[1], Pierre-Yves Vandenbussche[2], Colm J. Ryan[3], Walter Kolch[3,5,6], Dirk Fey[3,6]*

**1** Data Science Institute, National University of Ireland Galway, Ireland, **2** Fujitsu Ireland Ltd., Co. Dublin, Ireland, **3** Systems Biology Ireland, University College Dublin, Belfield, Dublin 4, Ireland, **4** Department of Oncology, Universidad de Navarra, Pamplona, Spain, **5** Conway Institute of Biomolecular & Biomedical Research, University College Dublin, Belfield, Dublin 4, Ireland, **6** School of Medicine, University College Dublin, Belfield, Dublin 4, Ireland, **7** Faculty of Informatics, Masaryk University, Brno, Czech Republic

†Lead author.
\* novacek@fi.muni.cz (VN); dirk.fey@ucd.ie (DF)

**Data Availability Statement:** The training/testing splits for reproducing the computational experiments are available at https://doi.org/10.6084/m9.figshare.12179925.v1. In case of queries

## Abstract

Phosphorylation of specific substrates by protein kinases is a key control mechanism for vital cell-fate decisions and other cellular processes. However, discovering specific kinase-substrate relationships is time-consuming and often rather serendipitous. Computational predictions alleviate these challenges, but the current approaches suffer from limitations like restricted kinome coverage and inaccuracy. They also typically utilise only local features without reflecting broader interaction context. To address these limitations, we have developed an alternative predictive model. It uses statistical relational learning on top of phosphorylation networks interpreted as knowledge graphs, a simple yet robust model for representing networked knowledge. Compared to a representative selection of six existing systems, our model has the highest kinome coverage and produces biologically valid high-confidence predictions not possible with the other tools. Specifically, we have experimentally validated predictions of previously unknown phosphorylations by the LATS1, AKT1, PKA and MST2 kinases in human. Thus, our tool is useful for focusing phosphoproteomic experiments, and facilitates the discovery of new phosphorylation reactions. Our model can be accessed publicly via an easy-to-use web interface (LinkPhinder).

## Author summary

LinkPhinder is a new approach to prediction of protein signalling networks based on kinase-substrate relationships that outperforms existing approaches. Phosphorylation networks govern virtually all fundamental biochemical processes in cells, and thus have moved into the centre of interest in biology, medicine and drug development. Fundamentally different from current approaches, LinkPhinder is inherently network-based and makes use of the most recent AI developments. We represent existing phosphorylation data as knowledge graphs, a format for large-scale and robust knowledge representation.

related to reagent and resource sharing, the point of contact is Systems Biology Ireland, University College Dublin (sbiadmin@ucd.ie).

**Funding:** This work was supported by the CLARIFY project funded by European Commission under the grant number 875160, the TOMOE project funded by Fujitsu Laboratories Ltd., Japan and Insight Centre for Data Analytics at National University of Ireland Galway (supported by the Science Foundation Ireland grant 12/RC/2289) and Science Foundation Ireland grants 14/IA/2395 and 15/CDA/3495 to Walter Kolch and David Matallanas, respectively. The funders had no role in study design, data collection and analysis, decision to publish or preparation of the manuscript.

**Competing interests:** The authors have declared that no competing interests exist.

Training a link prediction model on such a structure leads to novel, biologically valid phosphorylation network predictions that cannot be made with competing tools. Thus our new conceptual approach can lead to establishing a new niche of AI applications in computational biology.

## Introduction

Nearly all aspects of cell behaviour are controlled by phosphorylation events and intricate networks of kinases-substrate relationships mediating these phosphorylations [1]. Depending on the phosphorylation site, the attachment of a phosphate group can alter the activity of a substrate, its interaction with other proteins or its subcellular localization. This diversity of phosphorylation mediated processes control important cellular functions such as signal transduction, differentiation, migration, cell division and apoptosis. Dysregulation of these kinase-substrate relationships can have devastating consequences and are regularly observed in prevalent diseases, such as cancers or immune diseases. Therefore, kinases have emerged as attractive drug targets and have become the mainstay of targeted therapies with nearly fourty kinase inhibitors approved for clinical use as of 2018 [2] and over 150 in clinical trials since 2012 [3, 4].

In order to improve the design of kinase inhibitors, understand their mode of action and potential side effects, a better understanding of kinase-substrate relationships and the networks they form is necessary. With the advent of modern high-throughput mass-spectrometry based phosphoproteomics, many thousands of phosphorylation sites in substrate proteins can be identified [5]. However, large scale and reliable prediction of which kinase can phosphorylate which substrates at which sites remains challenging. High-throughput experiments are not informative in this case, because they cannot establish these detailed functional relationships, and addressing this issue in a one-by-one fashion is prohibitively expensive and time-consuming due to the large number of candidate interactions to be tested [6].

Reliable automated prediction of phosphorylation candidates is therefore much desired, because it can substantially reduce the number of possibilities that have to be tested experimentally. During the last decade, several tools for predicting phosphorylations have become available. The most widely used and recently described include: Scansite [7], GPS [8], NetPhos [9], NetPhorest [10], NetworKin [6, 10], PhosphoPredict [11]. Each of these tools, however, covers only a limited fraction of over 500 known human kinases [12], with 33, 217, 17, 178, and 6 kinases covered, accordingly. In addition to the limited coverage, existing approaches also suffer from an important conceptual limitation. Only intrinsic features of proteins (such as sequence, structure or functional annotations) are primarily used in training the predictive models. Phosphorylations, however, are inherent parts of complex interaction networks, and this type of information is largely neglected by current models.

Here, we show that predicting kinase-substrate relationships can be formulated as finding missing links in a knowledge graph (i.e. a relational, machine-readable knowledge base constructed from known phosphorylation networks). Knowledge graphs are a powerful way to organise descriptions of properties of objects and their connections [13]. However, they have not been widely used yet to analyse biological relationships. We show that using such a relational representation enables models that have superior generalisation power and precision when compared to existing approaches, lead to increased phosphoproteome coverage and produce biologically valid predictions. This can be explained by the fact that our approach fully utilises latent patterns in phosphorylation networks that are neglected by existing approaches

(e.g. long-range relational dependencies and implicit hierarchical structure). Moreover, the relational representation is not critically dependent on local features, which means our approach can make predictions even for under-researched proteins where existing approaches fail to provide results.

To test this concept, we have built a predictive model based the known phosphorylation network in PhosphoSitePlus [14] interpreted as knowledge graph. This model uses statistical relational learning to address the kinase-substrate prediction problem. We show that our model has superior predictive power based on a comparative validation trial following standard machine learning evaluatuion protocols. The model also outperforms existing tools in the total number of human kinases covered (327, nearly twice as many as the next best tool), which substantially increases the number of potential discoveries that can be made using our tool. The biological relevance of our approach is evidenced by the discovery and experimental validation of previously unknown kinase-substrate relationships for the AKT1, LATS1, PKA and MST2 kinases.

## Results

The concept of our approach in comparison with related existing techniques is illustrated in Fig 1 and details are given in the Materials and Methods section. Where existing tools use

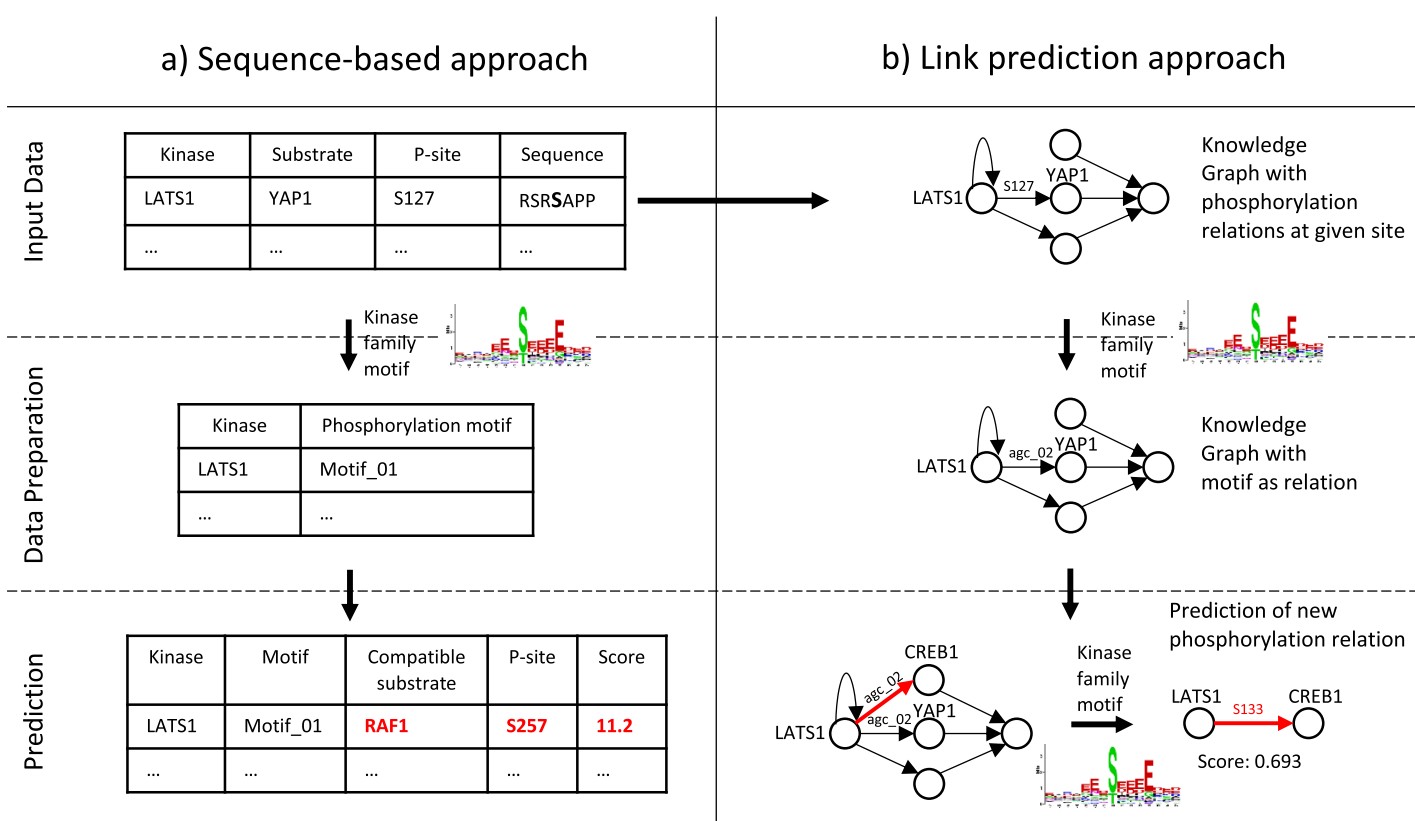

**Fig 1.** a) Sequence-based approaches aim to identify linear amino acid motifs that are phosphorylated by certain kinases. This is done based on known motif preferences of kinases, their groups or families. Each site and substrate is examined in isolation. Only limited numbers of well-studied kinases can typically be associated with substrates this way, and network context is largely ignored in such predictions. b) The LinkPhinder approach aims at learning regular patterns in a knowledge graph that represents the known kinase-substrate links as motif-based abstractions of the associated consensus sites. Based on the global, latent properties of the knowledge graph, the system can predict unknown, site-specific interactions between any kinase and substrate present in the input data.

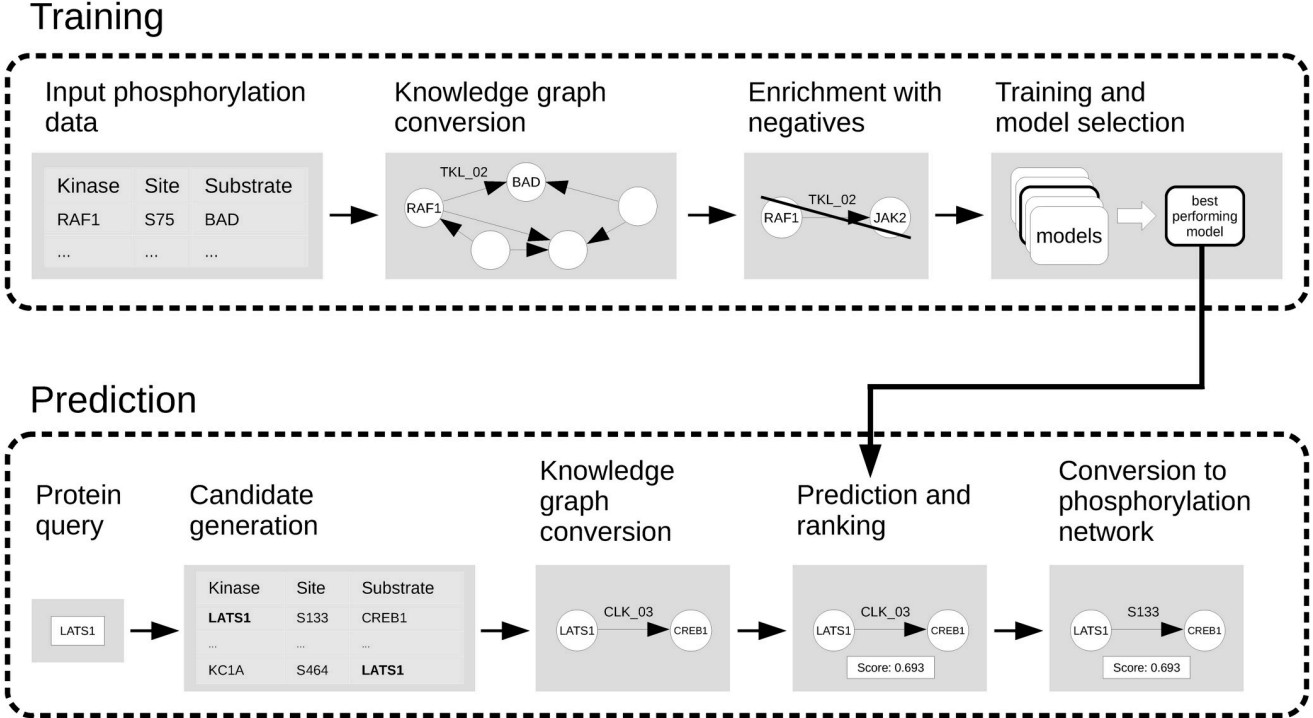

**Fig 2. The model is first trained on phosphorylation network data that has been converted to a knowledge graph representation.** Such a representation can be readily processed by link prediction algorithms (contrary to the original phosphorylation data). In the training stage, an optimal combination of model parameters is found and computationally validated. The optimal model is then trained on full phosphorylation network data and used for providing probabilistic ranking scores for all possible predictions that can be made using the input. Finally, reverse conversion technique is applied to the computed predictions to present them to users as residue-specific kinase-substrate relationships.

primarily local features based on sequence of particular proteins (left-hand side), our approach also considers the network information in training the model. Our predicitons are effectively based on explicit and implicit functional links between kinases and substrates represented as knowledge graphs. Briefly, we used a PhosphoSitePlus, a highly curated database of experimentally confirmed phosphorylation sites [14], to construct a knowledge graph where links between kinases and substrate corresponded to shared characteristics of kinase consensus sites. This knowledge graph represented a training set of known kinase-substrate relationships that was used for learning our predictive model (effectively, a multi-variate probability distribution function fitted to the input data). This model can consequently be used for predicting unknown kinase-substrate relationships with high coverage and precision.

The workflow of our methodology is illustrated in Fig 2. Details on training the computational model and the data used are provided in the Materials and Methods section. The main steps of constructing the LinkPhinder model are: (i) Generation of a phosphorylation network based on kinase-substrate pairs reported in PhosphoSitePlus (albeit any other database could be used). (ii) Inference of phosphorylation site motifs for kinase families based on quantifying the contribution of each amino acid in a set of consensus sequences to the likelihood that this sequence is phosphorylated. (iii) Conversion of the phosphorylation network into a knowledge graph using the phosphorylation motifs as generalised links to connect compatible kinase-substrate pairs while preserving the site information. (iv) Learning of new links based on both explicit and latent relationships in the input network data. The learning process is supervised and thus requires negative kinase-substrate relationships. These were generated using random

perturbations of the positive examples. (v) Selection of the best performing model. (vi) Generation of all possible kinase-substrate combinations using the input data and using our trained model for computing ranking scores for each kinase-substrate link. This ranking effectively allows to select most likely, previously unknown phosphorylations a kinase or substrate of interest can be involved in. (vii) Conversion of kinase-substrate links back to phosphorylation site sequences that provide the user exact information about the amino acid sequence phosphorylated by the given kinase in the substrate.

In the following, we present the performance of the LinkPhinder model. First, we benchmark LinkPhinder against six commonly used existing tools. Then, we present results of biological validation experiments focused on selected kinases of clinical relevance and their substrates. Finally, we introduce a web interface that allows the scientific community convenient access to LinkPhinder.

## Computational validation of LinkPhinder shows superior precision and kinome coverage

While the LinkPhinder model learns its parameters from the input data automatically, the optimal model configurations (also called hyperparameters) cannot be inferred that way and need to be determined empirically. In order to find these hyperparameters that optimise the performance of our model, we used the knowledge graph generated from PhosphoSitePlus [14] and evaluated several link prediction techniques across a range of their possible settings as described in the Materials and Methods section. The best method was ComplEx [15], which can handle large networks and generalises well for anti-symmetric relationships (of which the directed kinase-substrate links are an example). The optimal hyperparameters were identified by a grid search [16] and the best performing model was selected for the experiments described in this section. This model was trained on the entire network of phosphorylations contained in PhosphoSitePlus to produce unknown phosphorylation candidates for laboratory validation experiments described in the following sections.

The trained model can predict the likelihood of phosphorylation reactions that exist in the training dataset but have not been observed yet. In principle, any phosphorylation dataset can be used, but we chose PhosphoSitePlus because it is widely considered the most comprehensive and accurate dataset on known phosphorylations in many different organisms including human [17].

The computational validation experiments compared our approach to a selection of six existing and commonly used phosphorylation prediction techniques: Scansite [7], GPS [8], NetPhos [9], NetPhorest [10], NetworKin [6, 10] and PhosphoPredict [11]. For running this benchmarking trial, we generated 100 random train/test splits (90% train, 10% test) of true positives from the subset of PhosphoSitePlus human phosphorylations (i.e., *kinase-phosphorylation site-substrate* triples). A pool of negative statements was generated by random associations between all human kinases and *(phosphorylation site, substrate)* pairs available in PhosphoSitePlus. This pool was used for sampling as many negatives as there were positives in each train/test split. For each of the 100 splits, we trained our model on the 90% of the data and validated it on the unseen 10%. For the existing techniques, we generated all their predictions relevant to the proteins in the PhosphoSitePlus dataset and assessed them using the test splits.

Note that to make sure the presented relative differences between the methods are not merely due to the specific way we prepared the benchmarking data, we have also experimented with different train-test split and positive-negative ratios. The relative performances of the compared methods have not, however, changed from what is presented here. More

**Table 1. Comparative validation results.** AU-PR, AU-ROC refer to the area under the precision-recall and ROC curve, respectively. These metrics are widely used for validating predictive models based on ranking across their whole operating range [18]. P@K refers to the precision at K metric that gives the ratio of true positive statements ranked among top K results (e.g., P@10 refers to precision at 10; precision at 10 equal to 0.9 would mean that the corresponding tool typically returns 9 true positives among the top 10 results).

| Model | AU-PR | AU-ROC | P@10 | P@50 |
|---|---|---|---|---|
| GPS | 0.741±0.011 | 0.731±0.011 | 0.862±0.108 | 0.857±0.049 |
| NetworKin | 0.688±0.010 | 0.619±0.011 | 0.981±0.046 | 0.961±0.027 |
| NetPhorest | 0.650±0.012 | 0.598±0.011 | 0.905±0.091 | 0.905±0.041 |
| Scansite | 0.605±0.012 | 0.573±0.013 | 0.727±0.143 | 0.777±0.059 |
| Phosphopredict | 0.504±0.011 | 0.503±0.168 | 0.539±0.168 | 0.523±0.081 |
| Netphos | 0.612±0.012 | 0.563±0.013 | 0.865±0.105 | 0.863±0.048 |
| LinkPhinder | **0.973±0.004** | **0.968±0.004** | **0.994±0.024** | **0.993±0.012** |

information on the benchmarking methodology and results corresponding to the different ratios can be found in the Materials and Methods section.

The results are summarised in Table 1. The corresponding charts with the PR and ROC curves are given in Fig 3. The table presents means and standard deviations for each of the performance metrics computed across the 100 experimental runs with random train/test splits. Our model outperforms the existing techniques in all validated metrics, often by rather large margins. The narrower confidence margins of LinkPhinder results (about 1.8-42 times less than for the related works) mean that even if the experiment was done just once, it is still very likely the relative performance differences between the tools would be the same as presented in the table.

To gain additional insights into the presented results, we analysed to what extent each tool covered the phosphorylations in the test splits. The coverage is an important factor influencing the results since we assign zero scores to phosphorylations which the systems are not able to process (i.e., those for which no ranking scores can be produced). Therefore, a tool that does not produce scores for negative examples will have these annotated with zeros automatically and thus they will be at the bottom of their ranking lists. This is a possible advantage over tools that do produce scores for such negative phosphorylations, as any positive scores can only move the negatives up in the ranking, resulting in more false positive assignments.

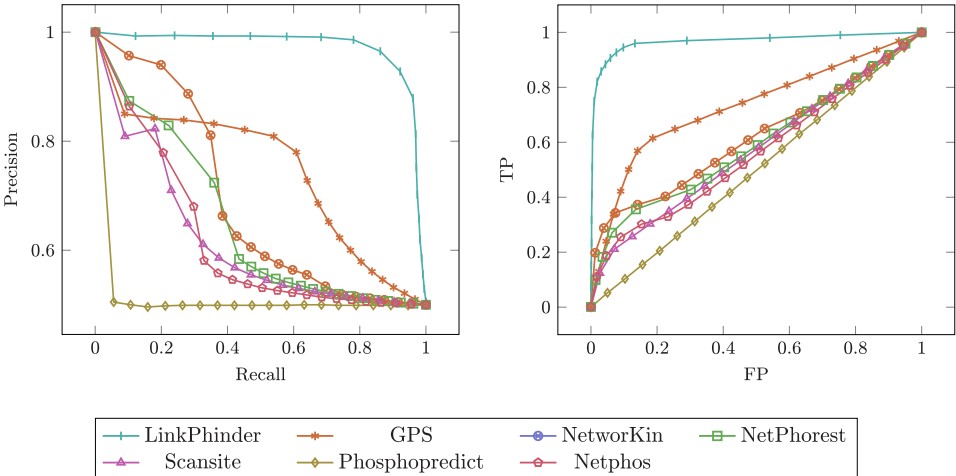

**Fig 3. The average precision-recall and ROC curves as per the experimental results reported in Table 1 (left and right part of the figure, respectively).**

**Table 2. Coverage of the tools in per cents.** Total, positive and negative coverage is given in the first three columns with data, respectively. The last column gives the percentage of missed negatives (i.e., negatives that are assigned the default zero score).

| Model | Tot. coverage | Pos. coverage | Neg. coverage | Missed neg. |
|---|---|---|---|---|
| GPS | 38.6±1.0% | 60.6±1.5% | 16.6±1.2% | 83.4% |
| NetworKin | 34.1±1.0% | 40.6±1.5% | 27.7±1.3% | 72.3% |
| NetPhorest | 34.1±1.0% | 40.6±1.5% | 27.7±1.3% | 72.3% |
| Scansite | 10.8±0.6% | 18.0±1.1% | 3.6±0.5% | 99.5% |
| Phosphopredict | 1.1±0.2% | 1.3±0.3% | 1.0±0.3% | 99.0% |
| Netphos | 28.8±1.1% | 33.0±1.7% | 24.7±1.2% | 75.3% |
| LinkPhinder | 64.2±0.8% | 97.0±0.5% | 31.4±1.5% | 68.6% |

The coverage of the different tools, i.e., their ability to make predictions for proteins represented in PhosphoSitePlus, is given in Table 2. This table shows that our model has the highest coverage of the test splits, especially when it comes to positive statements. However, the percentage of missed negative phosphorylations is still the lowest for our model, which means that the other tools are not disadvantaged by the setup of this benchmarking test.

Thus, LinkPhinder outperforms existing popular tools in terms of sensitivity and specificity (by means of the area-under-the-curve and precision metrics used), but also in terms of the number of predictions it can make. Importantly, LinkPhinder also covers a larger fraction of the human kinome than the other tools. Comprehensive visualisation of this fact is given in Figs 4 and 5.

The results diplayed in Fig 4 clearly illustrate the superior potential of LinkPhinder for discovering new phosphorylations relevant to under-researched kinases, which is currently considered one of the most pressing challenges in phosphoproteomics [17]. This is complemented by Fig 5 that shows, among other things, the relative advantages LinkPhinder presents in numbers of kinase-substrate and site-specific kinase-substrate interactions for which it can provide predictions (having the second best and best coverage, respectively). While higher coverage of possible predictions may not mean much on its own, we believe it is a reassuring sign when combined with the presented data on the superior performance of LinkPhinder in terms of the quality of the prediction scores it can associate with such an unprecedented range of kinase-substrate interaction candidates.

To provide a complementary computational validation using a dataset independent of the one we trained our model on, we have used a very recent data on site-specific interactions of 103 human kinases with their substrates in cancer cells [19]. Table 3 presents the performance of LinkPhinder and the six related tools when using this data for validation in the same fashion as in the previously reported computational experiment.

While the performance of all tools is substantially weaker than when using the PhosphoSitePlus benchmark (i.e. only slightly above the random baseline for both area-under-the-curve metrics), LinkPhinder is still the best in three out of four metrics, and close second in the remaining one. The overall poor performance can be attributed to a relatively small coverage of the [19] gold standard exhibited by most tools when compared to the PhosphoSitePlus [14] one (detailed overlap statistics are provided in section Training of the LinkPhinder Model). In such a situation, the relative ranks of the true positives among the rather large sets of all candidate predictions provided by the tools would tend to fluctuate quite widely, which can provide at least partial explanation of the differences in the predominantly ranking-based metrics between the two benchmark datasets. Another part of the explanation may be the fact that while [14] covers a broad range of cell lines and tissues, [19] only covers three cell lines. Tools that are presumably trained using existing knowledge covering as many cell/tissue types as

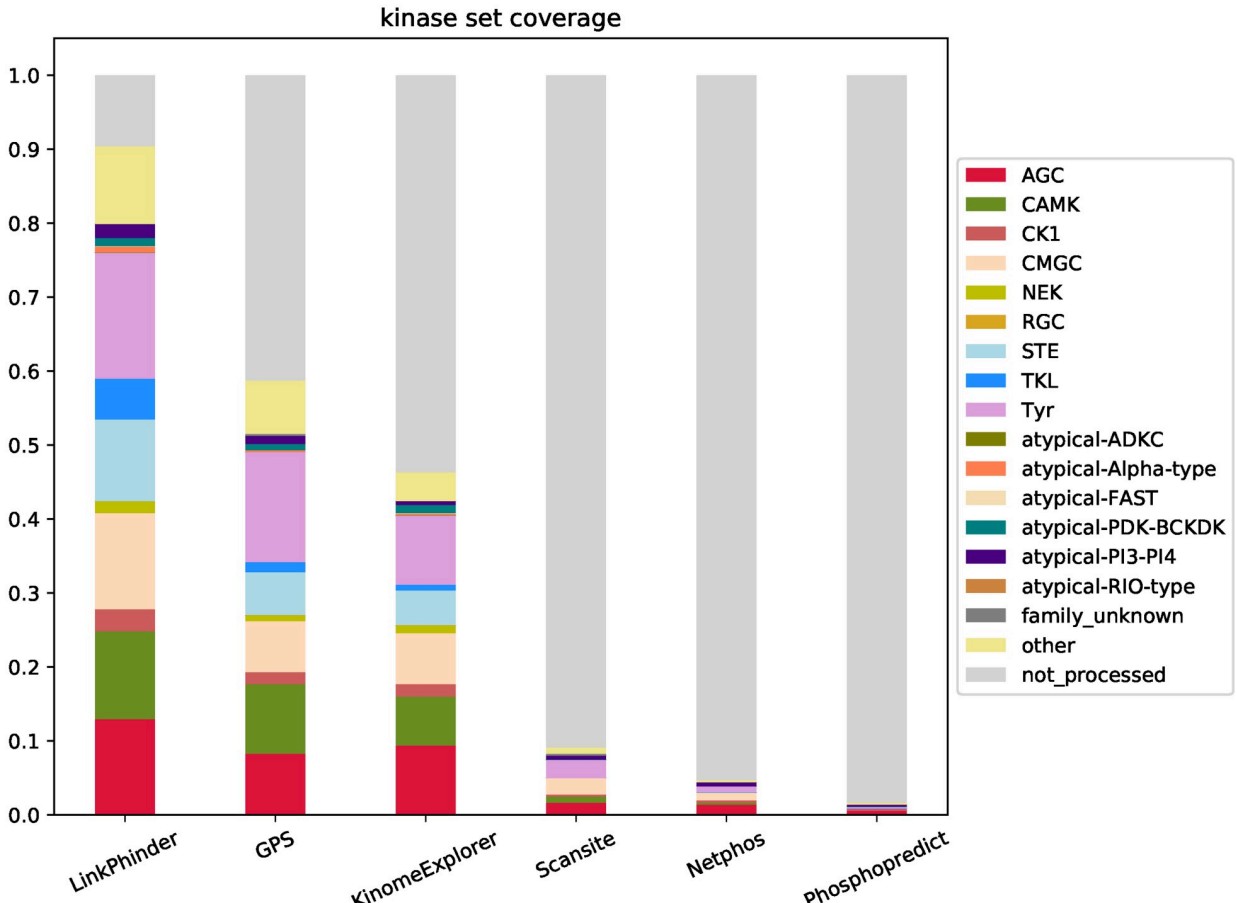

**Fig 4. Coverage of the human kinome and kinase families as per PhosphoSitePlus.** The "not_processed" category reflects the number of kinases for which a tool cannot produce any predictions. Note that NetPhorest and NetworKin only differ in scores assigned to predictions, while the set of phosphorylations they can produce scores for is identical. Therefore, they are grouped under a common KinomeExplorer [10] in the plot.

possible (like ours) may thus be expected to perform relatively poorly on a dataset that specifically covers a limited number of cell lines and corresponding kinases. That being said, this issue may point to an interesting research avenue to be addressed by future studies in this area that would further investigate the cell line-specific performance of models for predicting kinase-substrate interactions.

## Targeted experiments confirm two previously unknown phosphorylation sites targeted by LATS1 and AKT1

The rich dataset of over 11 million candidate predictions was assessed regarding its potential for discovering new phosphorylation sites for kinases that have biomedical relevance, such as AKT and LATS1. Both kinases regulate cell survival, growth, proliferation, and are frequently altered in cancer [20–22]. AKT has now become a leading drug target in cancer research, but the long term application of AKT inhibitors is still considered problematic because of AKT's essential roles in regulating glucose homeostasis [23]. The situation is similar with LATS1. Originally described as tumor suppressor, it also can have growth promoting roles [22]. In

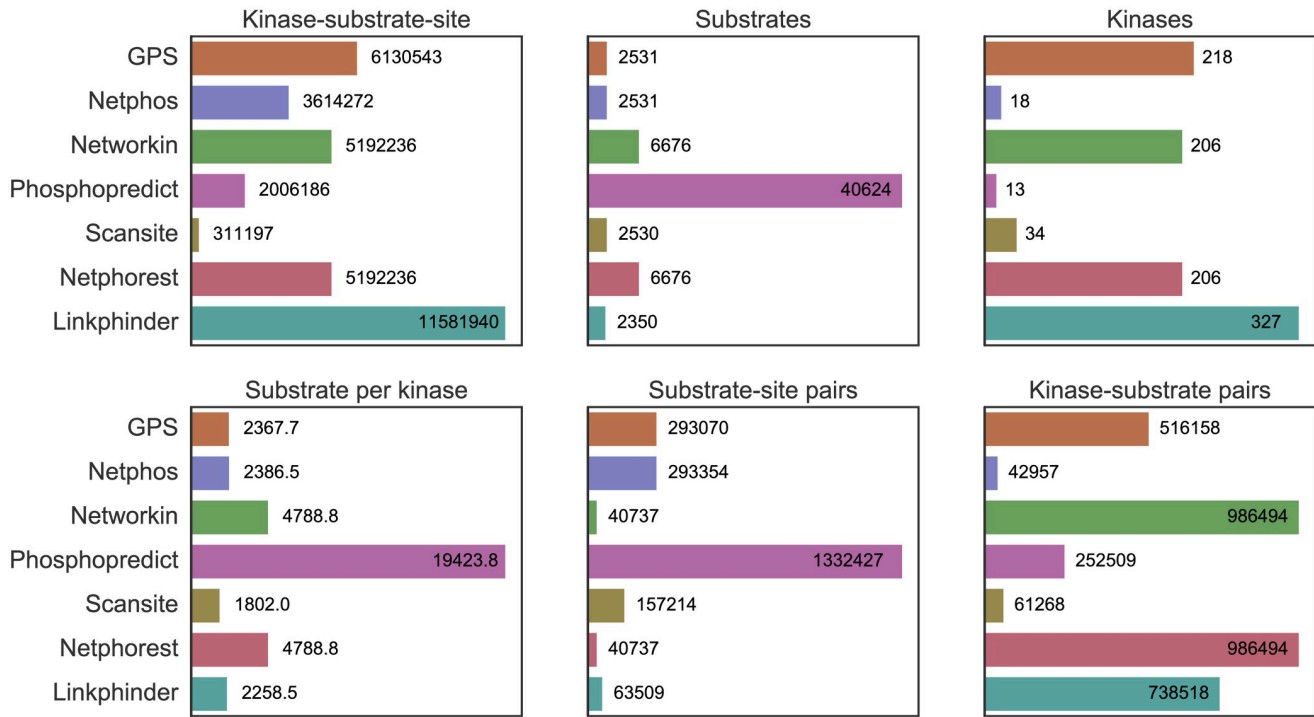

**Fig 5. Complementary statistics of the coverage of different systems in terms of number of kinases, substrates, sites per substrate, etc.**

order to resolve unwanted from desired effects a better and more comprehensive understanding of the substrate spectrum of these kinases is needed.

To this end, we have extracted high-stringency predictions for the LATS1 and AKT1 kinases. By visual inspection of this list, we were able immediately pinpoint several known and promising substrates of these kinases. YAP1 for example is the best characterized substrate of LATS1, and our tool predicted that LATS1 would phosphorylate YAP at serine 127, which is the best studied phosphorylation site that contributes to YAP inactivation [21]. Additionally, the list of the top 200 predictions for AKT contained eight bona-fide AKT substrates [24], for which several phosphorylation reactions were predicted. This means that our tool generated interesting, biologically relevant predictions.

Therefore, we decided to validate some of the new substrates experimentally. In order to select the most promising candidates we selected three proteins that are part of the wider LATS1 signaling network and for which antibodies were commercially available. To

**Table 3. Complementary computational validation of LinkPhinder using the recent dataset published in [19] as a benchmark independent of the primary training dataset (i.e. PhosphoSitePlus [14]).**

| Model | AUPR | AUROC | P@10 | P@50 |
|---|---|---|---|---|
| GPS | 0.518 ± 0.008 | 0.509 ± 0.010 | 0.675 ± 0.171 | 0.663 ± 0.059 |
| NetworKin | 0.519 ± 0.008 | 0.511 ± 0.010 | 0.682 ± 0.132 | 0.616 ± 0.062 |
| NetPhorest | 0.519 ± 0.007 | 0.510 ± 0.008 | **0.731 ± 0.135** | 0.659 ± 0.056 |
| Scansite | 0.504 ± 0.008 | 0.502 ± 0.009 | 0.561 ± 0.170 | 0.563 ± 0.066 |
| Phosphopredict | 0.502 ± 0.008 | 0.502 ± 0.009 | 0.519 ± 0.137 | 0.507 ± 0.069 |
| Netphos | 0.508 ± 0.009 | 0.505 ± 0.009 | 0.551 ± 0.149 | 0.554 ± 0.074 |
| LinkPhinder | **0.540 ± 0.009** | **0.532 ± 0.010** | 0.713 ± 0.153 | **0.671 ± 0.061** |

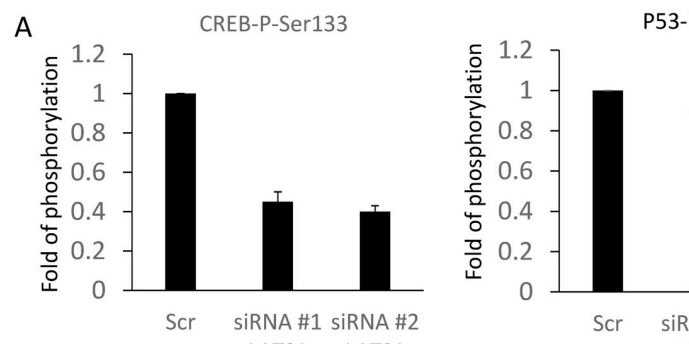

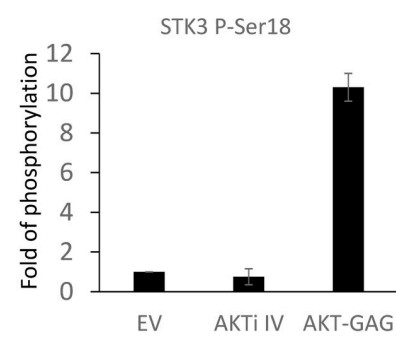

**Fig 6. Experimental validation of model predictions.** A) HEK293 cells were transfected with non targeted siRNA (Scr) of the indicated siRNA against LATS1. Phosphorylation of CREB or p53 was measured using specific antibodies and normalised to the level of expression of the corresponding proteins. The graph shows the fold change of the phosphorylation of the specific residues with respect to the Scr control. B) HEK293 were transfected with empty vector (EV) or GAG-AKT or treated with AKTi IV (10$\mu$M) for 1 hour. Phosphorylated proteins were immunoprecipitated using an anti-AKT antibody and the immunoprecipitates were blotted with anti-MST2. The bars show the fold change with respect to the control. The experiments were repeated at least 2 times. Error bars represent standard variations.

experimentally validate predicted LATS1 substrates we used the following strategy. HEK293 cells were transfected with two specific siRNAs against LATS1 in order to downregulate LATS1 protein levels (knockdown). Following this LATS1 knockdown, we would expect to see a decrease in the phosphorylation of LATS1 substrates. For confirmation, we used a positive control where we measured the phosphorylation of the known LATS1 substrate YAP1-S127, and indeed observed a decrease in YAP1-S127 phosphorylation in total cell lysates (c.f. S1A Fig). This control experiment demonstrated that our LATS1 knockdown works as expected and can be used to confirm potential LATS1 substrates.

One of the proteins that we selected for validation was CREB which is transcription factor that is regulated by phosphorylation [25]. This transcription factor is one of the best character-ized effectors of the MAPK ad PKA pathways. Evidence in the literature indicates that CREB modulates important LATS1 pathway functions by direct interaction with YAP1 and regula-tion of transcription [26]. Our tool predicted serine 133 of CREB (CREB-S133) as a putative substrate of LATS1. To confirm this, we used a specific antibody against CREB-S133, and saw that downregulation of LATS1 resulted in about 50% decrease of CREB-S133 phosphorylation (Fig 6, S1B Fig). This result clearly indicated that CREB is a physiological LATS1 substrate and highlights the potential of our tool to identify previously unknown kinase substrates.

In the case of AKT we decided to monitor putative substrates by manipulating the level of AKT activation using two strategies. Firstly, we inhibited endogenous AKT activity by using the specific chemical inhibitor AKTi IV. Secondly, we increased AKT activity by transfecting a kinase hyperactive form of AKT with gag-AKT [27]. One of the predicted AKT substrates is MST2 (MST2), which is an important protein kinase in the Hippo pathway that can phosphor-ylate and activate LATS1 [21], and which according to the prediction should be phosphory-lated at serine 18. Unfortunately, no commercially available antibody exists that could measure this phosphorylation site. Therefore, we employed an indirect approach to validate this predic-tion. We used an antibody that specifically binds to phosphorylated AKT substrates, which will immunoprecipitate (IP) all the proteins that are phosphorylated by AKT. Next, we blotted this IP using a specific antibody against MST2 (S1C Fig). The inhibition of AKT resulted in a slight, but consistent decrease of MST2 phosphorylation (0.75 fold), while expression of active AKT resulted in a 10 fold increase (Fig 6). The results validate the prediction that AKT1 phos-phorylates MST2.

After confirming the new site-specific kinase-substrate relationships involving the LATS1 and AKT1 kinases as reported above, we found out that none of the six existing systems used in our comparative validation could predict these phosphorylations on high stringency settings. This further demonstrates the unique power of LinkPhinder in the context of computational phosphorylation prediction.

## Mass spectrometry experiments confirm seven previously unknown phosphorylations by LATS1

To extend the targeted validation experiments we cross-referenced our predictions with high-throughput phospho-proteomic data on the LATS1 interactome (Fig 7A). This strategy is based on the fact that in order to phosphorylate a substrate, the kinase needs to bind to it. Based on our previous observations, kinases tend to be associated with their substrates in complexes that can be isolated and characterized by mass spectrometry [28]. Thus, by isolating all proteins that are bound to LATS1 by immunoprecipitation (IP) and analyzing this interactome using mass-spectrometry based proteomics, we should be able to identify a large number of LATS1 phosphorylation targets.

Using this approach, we obtained phospho-proteomic data on the LATS1 interactome from cells treated with two proapoptotic signals: FAS and etoposide, which both activate LATS kinase activity [29]. To identify the LATS1 interactome we transiently expressed GFP-LATS1 in HeLa cells, immunoprecipitated GFP-LATS1 with anti-GFP antibodies and identified the associated proteins using mass-spectrometry. Unspecific binding proteins were discarded by comparing with the control GFP IP. This approach identified seven proteins that were bound to LATS1 and phosphorylated on at least one residue (Fig 7B, S1 Data). These proteins are potential LATS1 substrates, but it is important to note that not all of these phosphoproteins are LATS1 targets, because LATS1 also binds to proteins that are phosphorylated by other kinases. Therefore, we cross-referenced this list of phosphorylated LATS1 interactors with our list of predicted LATS1 phosphorylation targets from LinkPhinder (Fig 7C, S1 Data). This confirmed 7 previously unknown phosphorylations on three substrates; five residues were phosphorylated on LATS1 (S613, S278, S464, S181, T17), one on MAP4 (S5), and one on ZMYM2 (T1253). Importantly, stimulation with FAS caused reduction of the phosphorylation of phosphorylation of LAST1-S464, MAP4-S5 and ZMYM2-T1253 (Fig 7D) indicating that regulation of these residues are specifpicaly regulated by the death receptor pro-apoptotic signal.

After confirming the new site-specific kinase-substrate relationships involving the LATS1 kinase as reported above, we searched for these in the prediction data provided by the existing tools. However, on high stringency settings, only GPS could predict one of the seven predictions made by us (LATS1-S464, GPS Score = 8.8, S2 Data). This further demonstrates the enhanced prediction capabilities of LinkPhinder.

## Kinase assays based on mass spectrometry confirm the sensitivity of LinkPhinder

One of the challenges to show the sensitivity of our tool and how it compares with existing tools is the lack of experimental methods to validate substrates systematically on a large scale. In order to further validate LinkPhinder predictions we decided to extend our validation experiments and use an in vitro kinase assay system that can identify multiple substrates for a given kinase. This method is based on the purification of proteins that have been phosphorylated by the kinase using an ATP analogue modified with a biotin group [30]. Briefly, all the endogenous kinases are inhibited with FSBA, a pan-kinase inhibitor, and the recombinant kinase is added to protein lysates together with ATP-biotin. ATP-biotin allows the purification

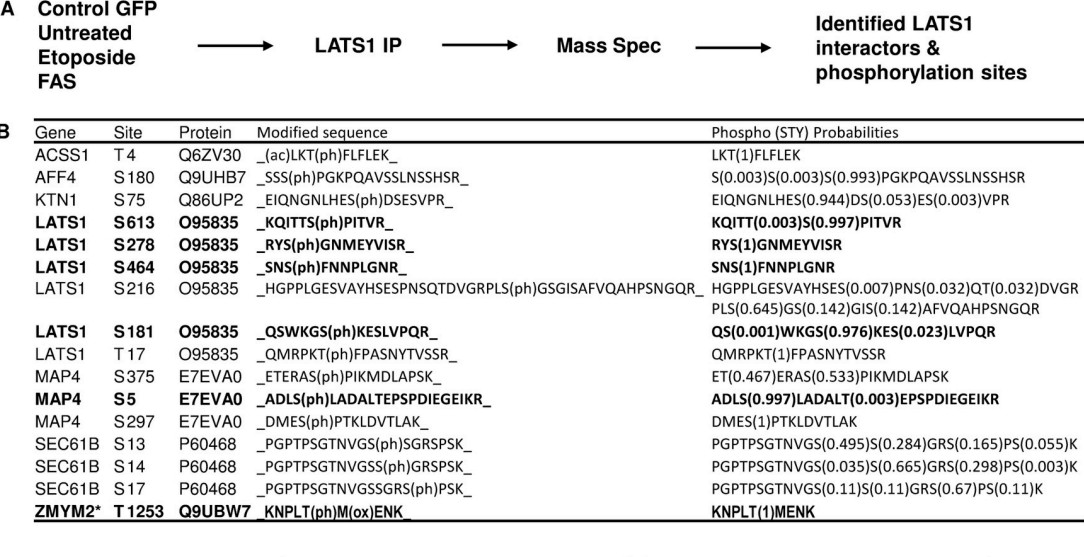

**A** Control GFP
Untreated → LATS1 IP → Mass Spec → Identified LATS1
Etoposide interactors &
FAS phosphorylation sites

**B**

| Gene | Site | Protein | Modified sequence | Phospho (STY) Probabilities |
|---|---|---|---|---|
| ACSS1 | T 4 | Q6ZV30 | _(ac)LKT(ph)FLFLEK_ | LKT(1)FLFLEK |
| AFF4 | S180 | Q9UHB7 | _SSS(ph)PGKPQAVSSLNSSHSR_ | S(0.003)S(0.003)S(0.993)PGKPQAVSSLNSSHSR |
| KTN1 | S75 | Q86UP2 | _EIQNGNLHES(ph)DSESVPR_ | EIQNGNLHES(0.944)DS(0.053)ES(0.003)VPR |
| **LATS1** | **S613** | **O95835** | **_KQITTS(ph)PITVR_** | **KQITT(0.003)S(0.997)PITVR** |
| **LATS1** | **S278** | **O95835** | **_RYS(ph)GNMEYVISR_** | **RYS(1)GNMEYVISR** |
| **LATS1** | **S464** | **O95835** | **_SNS(ph)FNNPLGNR_** | **SNS(1)FNNPLGNR** |
| LATS1 | S216 | O95835 | _HGPPLGESVAYHSESPNSQTDVGRPLS(ph)GSGISAFVQAHPSNGQR_ | HGPPLGESVAYHSES(0.007)PNS(0.032)QT(0.032)DVGR PLS(0.645)GS(0.142)GIS(0.142)AFVQAHPSNGQR |
| **LATS1** | **S181** | **O95835** | **_QSWKGS(ph)KESLVPQR_** | **QS(0.001)WKGS(0.976)KES(0.023)LVPQR** |
| LATS1 | T17 | O95835 | _QMRPKT(ph)FPASNYTVSSR_ | QMRPKT(1)FPASNYTVSSR |
| MAP4 | S375 | E7EVA0 | _ETERAS(ph)PIKMDLAPSK_ | ET(0.467)ERAS(0.533)PIKMDLAPSK |
| **MAP4** | **S5** | **E7EVA0** | **_ADLS(ph)LADALTEPSPDIEGEIKR_** | **ADLS(0.997)LADALT(0.003)EPSPDIEGEIKR** |
| MAP4 | S297 | E7EVA0 | _DMES(ph)PTKLDVTLAK_ | DMES(1)PTKLDVTLAK |
| SEC61B | S13 | P60468 | _PGPTPSGTNVGS(ph)SGRSPSK_ | PGPTPSGTNVGS(0.495)S(0.284)GRS(0.165)PS(0.055)K |
| SEC61B | S14 | P60468 | _PGPTPSGTNVGSS(ph)GRSPSK_ | PGPTPSGTNVGS(0.035)S(0.665)GRS(0.298)PS(0.003)K |
| SEC61B | S17 | P60468 | _PGPTPSGTNVGSSGRS(ph)PSK_ | PGPTPSGTNVGS(0.11)S(0.11)GRS(0.67)PS(0.11)K |
| **ZMYM2*** | **T 1253** | **Q9UBW7** | **_KNPLT(ph)M(ox)ENK_** | **KNPLT(1)MENK** |

**C** LinkPhinder predictions
for identified
MST1 interactors

| Kinase | Site | Substrate | Score | | Kinase | Site | Substrate | Score |
|---|---|---|---|---|---|---|---|---|
| LATS1 | S613 | LATS1 | 0.751187027 | | LATS1 | T17 | LATS1 | 0.751187027 |
| LATS1 | S278 | LATS1 | 0.751187027 | | LATS1 | S5 | MAP4 | 0.617851198 |
| LATS1 | S464 | LATS1 | 0.751187027 | | LATS1 | T1253 | ZMYM2 | 0.648494661 |
| LATS1 | S181 | LATS1 | 0.751187027 | | | | | |

**D**

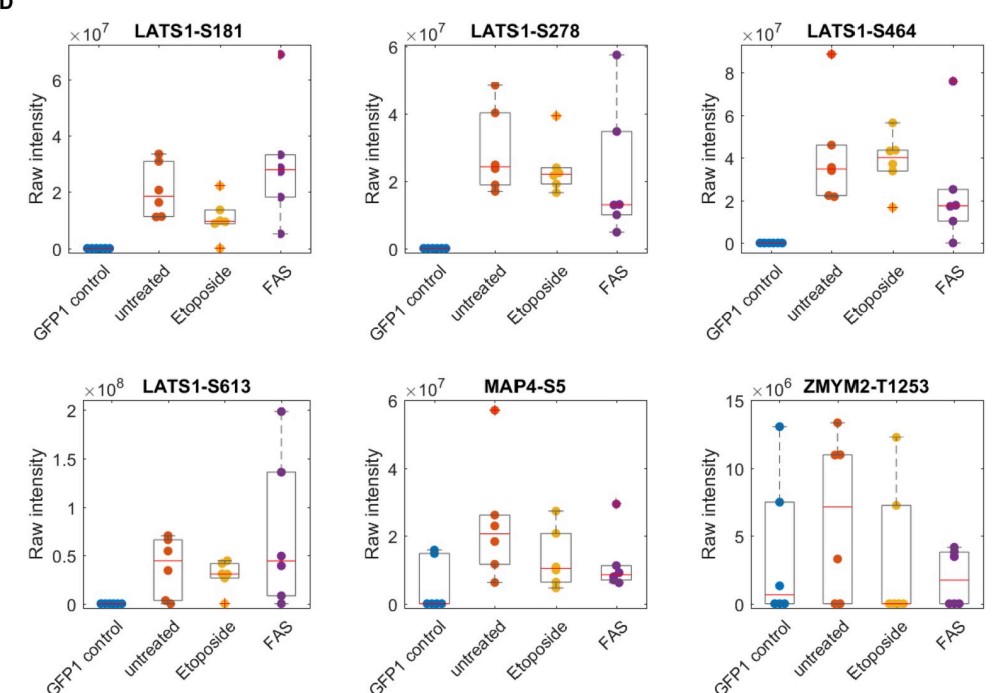

**Fig 7. Mass-spectrometry validation of a subset of LinkPhinder predicted phosphorylations.** A) Overview of the experimental design. B) Mass-spectrometry result: Specific LATS1 interactors and their phopshorylations. Bold rows indicate phosphorylation that were predicted by LinkPhinder. (*There is a risk that ZMYM2 binding might be unspecific. Some samples show high intensities in the GFP1 control, see panel D.) C) LinkPhinder predictions for the results in panel B. D) Mass-spec raw intensity values (dots) of the detected phosphorylation sites in GFP-LATS1 associated proteins under the indicated conditions (n = 6 replicates), and corresponding box plots indicating median (red line), upper and lower quartile (grey box), whiskers (most extreme values not defined as outliers), and outliers (plus marks) defined as values outside 1.5 times the interquartile range.

**Table 4. Sensitivity (*S*) of LinkPhinder substrate predictions per each of the kinase assay.**

| Kinase | Predicted substrate gene names | S |
|---|---|---|
| PKA | PKA, TGM2, PSMC5, PA2G4 | 0.57 |
| MST2 | MST2, MOB1A, NUP153, SNAPIN | 0.13 |
| LATS1 | LATS1, RHOA, VCP, SNAP25, CCT2, HNRNPK, RPS6, HSP90AA1 | 0.17 |

of phosphorylated proteins using streptavidin and the subsequent identification of these proteins as substrates using mass-spectrometry. In order to test the method we replicated the Pflum study using PKA as kinase in HeLA cells [30] in a different cell line (HEK-293). From a total of 834 identified proteins, 34 proteins were identified as putative substrates of PKA by comparing with the PKA deficient control samples (Table 4, and supplementary experimental information). Five of these proteins were previously identified in the Pflum study, and 11 of them were isoforms or proteins of the same protein family. We also identified 18 new putative substrates. These additional 18 proteins that did not occur in the Pflum study using HeLa cells may be cell-specific substrates in the here used HEK-293 cells. The overlap in the results clearly indicated that the global kinase assay is an additional tool that could be used to validate our predictions.

We then extended our validation experiments using this global kinase assay to LATS1 and MST2. First, we used LATS1 as kinase. We identified 240 putative LATS1 substrates from a total of 1397 identified proteins by comparing to the LATS1 deficient controls (Table 4, and detailed description in Section on Experimental Model and Subject Details). Secondly, we used MST2 as kinase. MST2 is another core kinases of the MST2/Hippo pathway with poorly characterised substrates. Our results identified 211 proteins as putative MST2 substrates. Strengthening our confidence into the validity of these results, five of the identified putative substrates have been described as MST2 interactors previously.

The experimentally validated PKA, MST2 and LATS1 substrate predictions made by LinkPhinder are listed in Table 4. The table also provides the sensitivity (*S*) of these predictions in the context of each specific kinase assay. The sensititivy was computed as

$$S = \frac{SUBS_{predicted}}{SUBS_{total}},$$

where $SUBS_{predicted}$ is the number of substrates for which LinkPhinder provided at least one site-specific phosphorylation prediction with a score above the high confidence threshold, and $SUBS_{total}$ is the number of substrates that were identified in the kinase assay and that are also present in the PhosphoSitePlus knowledge graph. Identified substrate proteins that were not in the knowledge graph were excluded for this analysis, because no predictions can be generated for those proteins.

The sensitivity of the PKA predictions was 0.57, which we consider a good result given that they were validated in an unbiased approach that has inherent technical limitations. For the poorly characterised MST2 and LATS1 kinases the sensitivities were lower, 0.13 and 0.17 respectively. It must be noted that generating predictions for MST2 and LATS1 is challenging because only a few substrates have been described experimentally, and most of the existing predictions tools could not generate predictions for MST2 and LATS1. Together these results indicate that LinkPhinder can be used to predict kinase-substrates interactions for poorly characterised kinases.

Finally, we wanted to benchmark LinkPhinder performance against the exsisting tools. However, we found this was not an easy task. Comparing LinkPhinder with existing tools

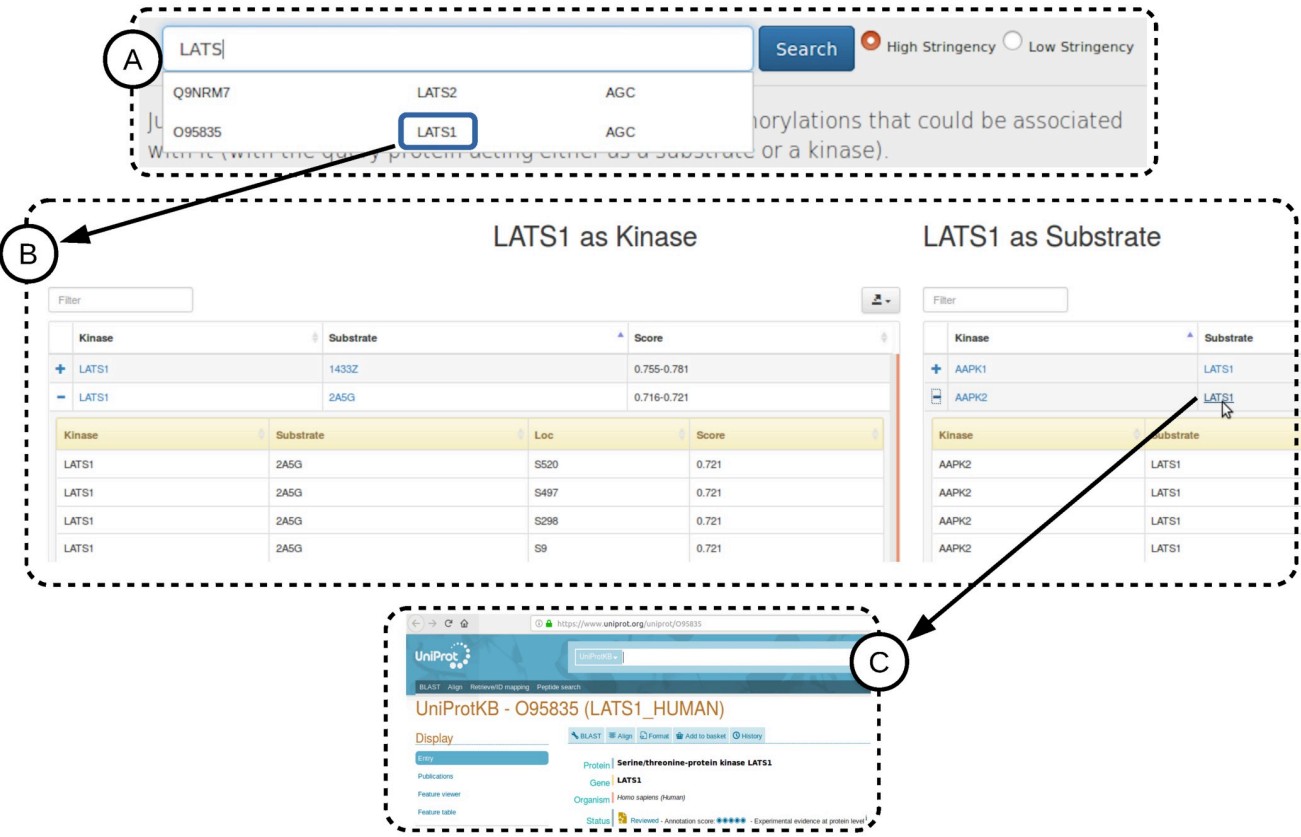

**Fig 8. The LinkPhinder web interface.** Shown is a typical search and browse interaction.

using the results of these experiments is not as straightforward as in the cases reported before. The main reasons are conceptually different methods for determining the decision threshold employed by each of the tools. This does not allow for direct comparisons in terms of sensitivity as defined above. However, one high-level observation can be made: Only GPS matches the coverage of LinkPhinder as it can produce predictions for all the three kinases we assayed. NetworKin and NetPhorest cannot compute any predictions for LATS1, NetPhos and Phospho-predict only cover PKA, and Scansite covers none of the assayed kinases.

## LinkPhinder web interface

In order to facilitate usage of LinkPhinder by the community we have developed an online interface available at https://LinkPhinder.insight-centre.org/.

A typical interaction with LinkPhinder is depicted in Fig 8. The corresponding instruction video is available in the *About* tab of the tool's web page. Briefly, the protein of interest can be entered into a search box with auto-completion (box A). Gene names and UniProt accession numbers are supported. The search is performed for high-stringency statements by default. However, all predicted statements can be searched as well (*cf.* the radio buttons in A). The query protein is evaluated by the system in two different ways, as a kinase and as a substrate and each type of predictions can be browsed independently (box B). The results can be filtered, and the predicted kinase-substrate pairs can be expanded to see the list of corresponding phosphorylation sites and prediction scores. Export of the predictions into a CSV file is also

possible. Further, users can easily access contextual information from a comprehensive protein database (UniProt) by clicking on the proteins in the results (box C).

## Discussion

In this work, we have overcome several limitations of the current phosphorylation prediction tools by representing phosphorylation networks as knowledge graphs. Knowledge graphs are a relatively new approach to representing relational knowledge in the Machine Learning, Artificial Intelligence and Semantic Web communities. They have quickly gained popularity for two main reasons. First, they can represent diverse types of knowledge in a simple format. Secondly, they are amenable to robust techniques of statistical relational machine learning, that can for example be used to discover new facts. The discovery naturally makes use of the entire structure of the knowledge graph (i.e. latent features and long range, implicit relationships instead of just local, explicit features). This makes the representation very useful in domains where complex network dependencies are critical. Kinase-substrate relationships are a good example of such a domain. Our results show that knowledge graphs enabled phosphorylation predictions that were not possible with existing tools that are primarily based on local features.

In particular, we have shown that phosphorylation networks can be meaningfully captured by knowledge graphs with kinases and substrate entities linked by relationships based on phosphorylation site motifs. Therewith, modern link prediction methods can be used to predict novel phosphorylation reactions and estimate their probability based on the entire network context. The resulting predictive model allows for making predictions about any protein present in the input data. This is a substantial advantage when compared with the existing tools. These tools typically focus on substrates as initial queries and include only a limited number of kinases. LinkPhinder not only covers a much broader range of possible kinase-substrate relationships than existing tools, but also shows very high generalization power and desirable ranking properties not exhibited by other, currently gold standard approaches. This aspect has been validated in experiments showing that our tool can generate numerous biologically valid predictions. Crucially, these predictions were not possible with a representative range of state-of-the-art tools (Scansite [7], GPS [8], NetPhos [9], NetPhorest [10], NetworKin [6, 10], PhosphoPredict [11]), demonstrating the utility of our tool.

More specifically, none of the LATS1 and AKT1 discoveries validated in targeted experiments were predicted with four out of six related tools starting with LATS1 or AKT1 as kinase queries. Only GPS and PhosphoPredict support such queries, but for less than 66.4% and 1.8% of the kinases covered by LinkPhinder, respectively. Furthermore, querying for the substrates directly did not predict any of the validated discoveries using any of the existing tools using their high stringency settings (if applicable; if controlling the stringency was not offered by a particular tool, we used all predictions made by the given tool). On medium stringency, the GPS tool could identify one prediction; the CREB1 phosphorylation by LATS1. On low stringency, the NetPhosK tool could also identify one prediction; the MST2 phosphorylation by AKT1. No existing tool could identify both predictions. The LATS1 predictions validated by the mass spectrometry experiments were not be predicted by any of the existing tools but one. Specifically, the GPS tool could predict one out of the seven predictions we made (LATS1 auto-phosphorylation at S464) on high stringency (and no further ones on lower stringencies). The other five tools could not identify any of our validated predictions. When cross-referencing the list of LATS1 predictions from other tools with our predictions, no additional predictions were made, demonstrating that our tool has the best coverage. Together, these results clearly illustrate the advantages of our tool.

Experimental validation using the global PKA, MST2 and LATS1 kinase assays showed promising results in terms of LinkPhinder's sensitivity for identifying new substrates. Direct comparison with the existing tools was not possible due to disparate methods employed by each tool in determining their decision/high-stringency threshold. However, the results reconfirmed one significant benefit of LinkPhinder. We were able to produce substrate predictions for all three kinases studied, which was not possible with five of the six existing tools, with the exception of GPS, again demonstrating that LinkPhinder's increased kinase covarage is an important contribution.

To build on the work presented here, we intend to incorporate more contextual data (e.g., relevant protein interactions from STRING or pathway data from Reactome) to see whether they can bring new and/or more accurate predictions pertinent to clinically relevant pathways. We also want to develop predictive models that would utilize the biology of phosphorylation directly in the training process and not only in the knowledge graph conversion and negative example generation. As demonstrated, incorporating more network context and biological knowledge into the prediction process has great potential to further increase the coverage, predictive power, and usefulness of the resulting tools.

Another research direction to explore in future is the applicability of our predictive model to improving the accuracy and scope of methods for predicting downstream effects of kinase signalling or the kinase activity profiles. An example of such method that could benefit from our results is described in [31]. We believe follow-up experiments combining focused phosphoproteomics studies like this with our model will further demonstrate the practical relevance of the work presented here.

## Materials and methods

### Computational model and validation details

**Datasets and tools used.**   To compile the phosphorylation network that is the primary input for building the LinkPhinder model, we used the PhosphoSitePlus dataset in a version available on 26th of June 2017 (c.f. https://www.phosphosite.org/staticDownloads.action). There were 10,173 phosphorylation statements on 362, 7,302 and 2,377 distinct kinases, substrate-site combinations and substrates in the compiled phosphorylation network, respectively. Note that in the construction of all datasets, we have focused only on the *Homo Sapiens* species, unless specified otherwise.

In order to convert the phosphorylation statements extracted from PhosphoSitePlus into a knowledge graph, we had to compute motifs characteristic to the context sequences of phosphorylation sites. For that task, we used the MEME tool, version 4.11.2 (c.f. http://meme-suite.org/doc/download.html?man_type=web).

We used three state of the art knowledge graph embedding and link prediction methods to train a model that can discover new links in the phosphorylation knowledge graph. The methods are TransE [32], DistMult [33] and ComplEx [15].

The PhosphoSitePlus dataset, together with UniProt (c.f. http://www.uniprot.org/) was also used for generating a mapping between substrates and their possible phosphorylation sites. This mapping was used in the conversion of the internal, motif-based knowledge graph statements to phosphorylation statements when computing scores of possible phosphorylations that have not been known before. We focused only on substrates present in our knowledge graph, which resulted in 74,142 distinct substrate-site pairs that can be used for generating candidate phosphorylations (i.e. potential discoveries).

To assess LinkPhinder in comparison with related state of the art systems, we downloaded and/or generated full sets of phosphorylation predictions that can be made with the following

tools: Scansite 3 (c.f. http://scansite3.mit.edu), KinomeExplorer (predictions produced by two tools, NetworKIN and NetPhosK, c.f. http://kinomexplorer.info/), Netphos (c.f. http://www.cbs.dtu.dk/services/NetPhos/), GPS (c.f. http://gps.biocuckoo.org/index.php) and Phospho-Predict (c.f. http://phosphopredict.erc.monash.edu/). The numbers of predictions that can be made with the corresponding tools are as follows: 6,130,542 (GPS), 5,192,235 (KinomeEx-plorer), 3,614,271 (Netphos), 2,006,185 (PhosphoPredict), 311,196 (Scansite 3). The numbers of high-stringency predictions are not straightforward to determine using the set of all predictions available, since some tools allow for stringency settings just at the level of manual, single-protein queries. Thus we were only able to establish the number of high-stringency predictions for Scansite, NetPhos and PhosphoPredict: 12,346, 212,107 and 132, respectively.

**Construction of the phosphorylation network and knowledge graph for training the model.**   The construction of the phosphorylation network requires data sources containing relation information of kinase, substrate and substrate's amino acid phosphorylation site. In our experiment, we used PhosphoSitePlus kinase-substrate dataset, an experimentally determined substrates, sequences, cognate kinases, and metadata curated from the literature [14]. Only relations involving a kinase and substrate protein for the human species were considered (*KIN_ORGANISM == SUB_ORGANISM == 'human'*). Although the dataset includes phosphorylation site's amino acids context sequence of size 7, we did not use that information as we wanted to experiment with different and potentially larger context sequence sizes. Instead we extract the context sequence from UniProt (Universal Protein Resource) and more specifically from the reviewed (Swiss-Prot) main protein sequence (*uniprot_sprot.fasta*) and from isoform sequences (*uniprot_sprot_varsplic.fasta*). We discard any relation in the kinase-substrate dataset for which the phosphorylation site does not match the UniProt sequence. Table 5 presents some statistics about the phosphorylation network.

The knowledge graph conversion makes use of kinase family consensus motifs to transform phosphorylation network statements to knowledge graph relations. The kinase families classification is extracted from UniProt's human and mouse protein kinases: classification and index. Only information about human kinases which are part of the phosphorylation network are kept.

The conversion of phosphorylation network data into knowledge made use of the MEME tool in a pipeline graphically described in Fig 9.

To realise the step 3 of the above pipeline we used specifically the `meme` command line utility for sequence motif discovery, version 4.11.2. MEME was applied in parallel on batches of site context sequences drawn from substrates targeted by kinases of the same family. The size of the batches was a configurable hyper-parameter of the conversion and model training process. We used values ranging over the set {50, 100}. The static parameters used for every invocation of the MEME tool were: `-text`, `-protein`, `-mod zoops`, `-x_branch`, `-minw 2`.

**Table 5. Phosphorylation network components statistics.**

| No. of elements in the phosphorylation network | |
| --- | --- |
| Phosphorylation relations | 9,802 |
| Kinases | 327 |
| Substrates | 2,350 |
| Phosphorylation sites | 7,083 |
| Avg. No. of substrate/kinase | 7.19 |
| Avg. No. of substrate's site/kinase | 21.66 |

**Fig 9. High-level workflow of generating predicate labels for the phosphorylation knowledge graph based on motifs extracted from the context sequences of phosphorylation sites by means of the MEME tool.**

The MEME parameters that were dependent on the specific properties of the sequence batch and/or hyper-parameters of the whole model were: `'-maxw MW, -maxsize MS, -nmotifs NM, -bfile BF` where `MW` was the maximum width of a sequence in the batch, `MS` was the maximum width multiplied by the number of sequences in the batch, `NM` was the maximum number of motifs to be generated (set conservatively to 10 in the reported experiments as no batch generated more motifs than that number under any tested settings) and `BF` was a background Markov model of order 5 generated from the sequence batch.

Table 6 presents some statistics about the generated knowledge graph.

**Training of the LinkPhinder model.   Generating a phosphorylation knowledge graph.** Before we could train a statistical relational learning model, we had to construct a knowledge graph representing the known phosphorylation information. As the primary input into the knowledge graph, we chose a phosphorylation network compiled from the PhosphoSitePlus [14] data set (focusing on *Homo Sapiens* species only). In principle, any phosphorylation data can be used, but PhosphoSitePlus is well curated and comprehensive making it an ideal starting point. There were 10,173 site-specific phosphorylation statements on 363 and 2,377 distinct kinases and substrates, respectively, in the compiled phosphorylation network. The network consists of statements $\langle K, L, S \rangle$ where $K, L, S$ are kinase, phosphorylation site and substrate, respectively. The biological meaning of such statements is that the kinase $K$ phosphorylates the substrate $S$ by binding to it and attaching a phosphoryl group to the site $L$.

To convert the phosphorylation network into a knowledge graph, we utilised motifs of phosphorylation sites preferred by specific kinase families. For each kinase family as defined in [34], we computed a set of consensus sequence motifs using the MEME tool run with parameters described in the previous section. The input to the tool were sets of sequences representing the local context of $2k + 1$ amino acids surrounding all phosphorylation sites in substrates targeted by the kinases in each family. The value of $k$ was a configurable hyperparameter of the conversion algorithm representing the context size, i.e. the number of amino acids on the left and right side of the phosphorylation site. See section on Finding the Optimal Hyperparameters of the Model for details on the other hyperparameters. The output of the conversion process were motifs that characterise the local context of the kinase-substrate interaction using

**Table 6. Knowledge graph components statistics.**

| No. of elements in the knowledge graph | |
|---|---|
| Motif-based relations | 9,956 |
| Kinase families | 12 |
| Kinase family motifs (relation types) | 24 |
| Avg. No. of motif/kinase family | 2.00 |

a position-specific scoring matrix, that quantifies the relative contribution of each amino acid in the substrate sequence. The scoring matrices were extracted from the text output of the MEME tool executed as described above. The motifs were consequently used for converting the $\langle K, L, S \rangle$ statements coming from the input phosphorylation network to labeled knowledge graph edges $\langle K, M, S \rangle$, where $M$ is a link label (also called a typed relation) that corresponds to a motif compatible with the family of $K$ and the site $L$ in the substrate $S$. Here, compatibility means a positive score of the site's context sequence with respect to the position-specific scoring matrix of the motif $M$.

The end result of the conversion is a knowledge graph consisting of true positive statements $\langle K, M, S \rangle$. Here, a protein may act as kinase in several statements and as substrate in several other statements. Therewith, these statements describe entire known phosphorylation network from PhosphoSitePlus.

**Generating negative statements based on the phosphorylation biology**. The knowledge graph generated from the phosphorylation network can be used for discovering new kinase-substrate relationships by means of link prediction [35], which is a technique for estimating likelihood of existence of a typed relationship between two entities based on other observed relationships in the data. The typical intention is discovering new relations that are not explicitly present in a knowledge graph. Training a link prediction model is a supervised machine learning process, and therefore requires negative examples in addition to the positive statements in the phosphorylation knowledge graph. Such negative examples are typically created by corruption of the positive statements by introducing random entities as part of the positive relation statements [35]. In our case, this technique could lead to correct kinase-substrate relationships being treated as negatives because kinases are promiscuous (i.e. one kinase can phosphorylate many substrates and one phosphorylation site can be targeted by many kinases). Hence, random corruptions of true statements may generate many false negative statements. Such false negatives would adversely affect the discriminative power of the model. Therefore, we need to impose specific restrictions when generating negative statements. We based these constraints on biological knowledge as follows. Firstly, most kinases belong to families that usually share substrates, while different families tend to phosphorylate different substrates [34]. Secondly, substrates are unlikely to be phosphorylated by a kinase if they have highly incompatible phosphorylation sites with respect to the kinase consensus motif. This incompatibility directly motivates two types of corruptions. For a statement $\langle K, M, S \rangle$, valid corruptions are: i) statements $\langle \bar{K}, M, S \rangle$ such that $\bar{K}$ is from a different family than $K$; ii) statements $\langle K, M, \bar{S} \rangle$ such that all phosphorylation sites in $\bar{S}$ score negatively with respect to the scoring matrix of the motif $M$.

**Training the model on the full input dataset to maximise its generalisation power**. The model with best-performing hyper-parameters was retrained on the entire knowledge graph derived from PhosphoSitePlus. This is appropriate due to the excellent numerical stability reported in Table 1. The main reason for training the model on the entire dataset is that such a strategy is preferable for making new discoveries because it uses all available information.

The model can be used for computing probabilistic ranking scores (with values between 0 and 1) of predictions ranging across all possible combinations of kinases, sites and substrates present in PhosphoSitePlus, and thus contribute to the discovery of previously unknown phosphorylations.

As described in Fig 2 and the prior parts of this section, the core link prediction model works on the converted knowledge graph, which means that it can only deal with relationships that abstract the site information using motifs. Putative phosphorylations for which the model is supposed to compute scores, therefore, have to be converted to the same form. After the

converted phosphorylation statements are scored, they have to be transformed back to the form that contains the specific phosphorylation site. This conversion is dual to the knowledge graph conversion—each statement $\langle K, M, S \rangle$ corresponds to statements $\langle K, L, S \rangle$ such that $L$ is a known phosphorylation site in $S$ (as per the PhosphoSitePlus [36] and UniProt data sets) that scores positively with respect to the position-specific scoring matrix of the motif $M$.

Given a single protein as a query, the model can produce a ranked set of candidate phosphorylation sites that involve the protein either as a substrate or as a kinase. The ranked list can optionally be filtered using high- or medium-stringency thresholds. We apply a threshold derived from the manually curated phosphorylation network we use as an input—the high-stringency threshold is a value such that 99.5% of the known phosphorylations score above it (the value is 0.672 in the reported model). The medium-stringency threshold is 0.5 (i.e. a score that indicates higher-than-random plausibility of the given statement). The ranking of the results reflects the global network context of all known phosphorylation sites and kinase-substrate relationships represented in the input knowledge graph, which is a type of information that is not incorporated by any other existing tool. Moreover, the predictions can be generated on any protein, be it a kinase or substrate.

This coverage and flexibility makes our model more powerful than most existing phosphorylation prediction tools that can only be queried for substrate proteins (in the GPS and PhosPredict tools, one can generate predictions associated with a kinase, but the systems combined still cover only about half of the kinases covered by LinkPhinder).

In total, LinkPhinder can produce 11,581,940 predictions when applied to all putative phosphorylations that can be generated from the proteins and phosphorylation sites present in the input data (PhosphoSitePlus). Out of these, 2,009,171 and 7,232,636 are of high and medium stringency, respectively. We can make predictions for 327 human kinases, nearly twice as many predictions than the next best among six related methods we have tested (GPS [8], with 217). This shows substantial improvement in the kinome and also general proteome coverage.

Further details and information about the coverage of LinkPhinder compared to other systems can be found in Table 7.

**Finding the optimal hyperparameters of the model**. Prediction of phosphorylation reactions is based on models trained on the knowledge graph data consisting of positive and negative statements. Negative statements are computed via perturbation of positive statements by means of ad-hoc operators. In our experiments, two negative statements are generated from each positive statement. Data is split into training+validation and testing. In particular, eighty percent of the available data is used for training and validating the models and the remaining part is used for testing. This data is used to evaluate multiple link prediction techniques with the aim of optimising prediction performance. For each of these, a grid search within the space

**Table 7. Statistics of the coverage of the different predictive systems and their overlap with the [19] gold standard.** The letters S and K in the column headers denote substrates and kinases respectively.

| Model | Triplets | Kinases | Substrates | K-S pairs | S-S pairs | S per K | Sites per S |
|---|---|---|---|---|---|---|---|
| Cutilass20 | 19066 (100.0%) | 103 (100.0%) | 2556 (100.0%) | 15178 (100.0%) | 6090 (100.0%) | 147.4 | 2.4 |
| GPS | 6130543 (5.3%) | 218 (62.1%) | 2531 (35.9%) | 516158 (23.7%) | 293070 (42.8%) | 2367.7 | 115.8 |
| Netphos | 3614272 (2.8%) | 18 (5.8%) | 2531 (35.9%) | 42957 (2.7%) | 293354 (43.2%) | 2386.5 | 115.9 |
| Networkin | 5192236 (0.0%) | 206 (55.3%) | 6676 (70.0%) | 986494 (35.1%) | 40737 (0.0%) | 4788.8 | 6.1 |
| Phosphopredict | 2006186 (0.0%) | 13 (1.0%) | 40624 (99.7%) | 252509 (0.1%) | 1332427 (25.1%) | 19423.8 | 32.8 |
| Scansite | 311197 (0.7%) | 34 (16.5%) | 2530 (35.9%) | 61268 (5.3%) | 157214 (36.5%) | 1802.0 | 62.1 |
| netphorest | 5192236 (0.0%) | 206 (55.3%) | 6676 (70.0%) | 986494 (35.1%) | 40737 (0.0%) | 4788.8 | 6.1 |
| LinkPhinder | 11581940 (26.2%) | 327 (84.5%) | 2350 (33.2%) | 738518 (35.7%) | 63509 (39.7%) | 2258.5 | 27.0 |

of available hyperparameter values of the models is performed. For each configuration, 10-fold cross validation is run. The combination of prediction technique and parameters that delivers the best performance is selected and this information is used to train a model on all the available data in order to exploit the entire knowledge about the phosphorylation reactions that have been experimentally validated.

Three link prediction techniques have been used, they are: *TransE*, *DistMult* and *ComplEx* [15, 32, 33]. TransE is one of the earliest techniques to have been proposed and its simplicity makes it a valid reference to learn about embeddings. In our case these embeddings are entities and relation types that are represented by means of vectors of the same length. A true statement is expected to satisfy the vectorial expression *subject + relation type* ≈ *object*. DistMult adopts a different approach, the score is the sum of the element-wise products between the subject vector, a diagonal matrix representing the relation type and the object vector:

$$score = \sum_{i=1}^{d} subject_i \cdot relation_i \cdot object_i$$

This denotes that the score is not built considering inter-relations between different latent features. ComplEx follows the same approach as DistMult, with the difference that complex numbers are used in place of real values. The score is the real part of the score formula used in DistMult.

The hyperparameters that control model generation are, in this order: number of negatives generated for each positive statement; number of training epochs through which the model parameters are optimised; number of batches in which data for model training is divided; batch size of amino acid sequences for motif generation (it affects the number of relation types); number of dimensions of vectors; margin of the hinge loss; distance function for computing similarity (only for TransE); learning rate of the model and, ultimately, context size, namely, the number of amino acids to consider on the left and on the right of the binding site. While for some hyperparameters values are selected from a set, for others the values are fixed as they were determined by means of independent experiments. Their respective values are listed in Table 8.

The link prediction technique that delivers the best performance is ComplEx with vectors of size 50 and context size equal to 15. This configuration was used to train a model on the entire network of phosphorylations and their associated negatives. The trained model is used to predict the likelihood of unobserved phosphorylation reactions actually existing in nature.

**Table 8. Hyperparameters space used by grid search to identify the best model ($L_1$, $L_2$ stand for Manhattan and Euclidean distance norms, respectively).**

| hyperparameter | values |
| --- | --- |
| number of negatives | 2 |
| number of epochs | 100 |
| number of batches for model training | 10 |
| batch size for motif generation | 50 |
| embedding size | {50, 100, 150, 250, 500} |
| margin | 1 |
| similarity (only TransE) | {$L_1$, $L_2$} |
| learning rate | 0.1 |
| context size | {7, 15} |

**Construction of the state of the art prediction data sets.**   The following paragraphs describe the construction of sets of predictions computed by existing tools that are used in comparative validation of the LinkPhinder model.

**Scansite 3**. Scansite searches for motifs within protein substrates that are likely to be phosphorylated by a specific protein kinase. It takes as input a protein substrate ID and sequence and gives as output a confidence score for given substrate amino acid sites to be phosphorylated by one of 70 kinases handled by the system. We queried the system with all substrates contained in our phosphorylation network and separately accepted results with low and high stringency leveld.

**NetworKIN and NetPhorest**. KinomeXplorer framework contains results of both NetworKIN and NetPhorest systems with only the score changing. The KinomeXplorer dataset uses gene identifiers to refer to protein phosphorylation. In order to compare the results with the validation set we had first to use UniProt gene query to recover the protein identifier. After downloading the dataset, we queried UniProt using both EmbID and gene name to resolve a protein ID. In case a query did not yield any result or multiple proteins were returned, the original statement was omitted. Finally, we kept only the system protein identifier-based statement responses that pertained to the proteins contained in our phosphorylation network.

**NetPhos 3.1**. The NetPhos 3.1 system predicts serine, threonine or tyrosine phosphorylation sites in eukaryotic proteins using ensembles of neural networks. The system can provide predictions for 17 kinases only. Using the stand-alone software package, we queried the system with all substrates and associated sequences contained in our phosphorylation network. The results obtained ar low and high stringency levels were used seperately.

**GPS 3.0**. Group-based Prediction System (GPS) predicts phosphorylation sites with their cognate protein kinases using a four level kinase hierarchical structure in multiple species. We used the batch predictor of the desktop application to pull out results for all substrates and associated sequences contained in our phosphorylation network.

**PhosphoPredict**. The PhosphoPredict system reportedly predicts kinase-specific substrates and the corresponding phosphorylation sites for 12 human kinases, including CSNK1A1, CSNK2A1, PRKACA, ATM, AKT1 (aka. PKB), SRC, GRK, PKC, GSK, CaMK, CDKs and MAPKs. However, only six of these actually correspond to single kinases, whereas the other seven are often rather diverse families of different proteins (CDKs, MAPKs, PKC, GRK, GSK, CaMK), and thus we focused on them in our comparison. PhosphoPredict employs a feature selection method based on the minimum Redundancy and Maximum Relevance (mRMR) to select the most informative feature subsets that contribute to the prediction success of each kinase families. We kept only those system statements which referred to the proteins present in our phosphorylation network.

**Comparative computational validation.**   A comparative evaluation was performed with the purpose of assessing the performance of LinkPhinder in the context of existing phosphorylation prediction methods (i.e. GPS, NetworKin, NetPhorest, NetPhosK, Scansite and Phosphopredict). Since the process of training LinkPhinder is stochastic, the performance changes slightly every time a new model is trained. To minimise the variability of the results, and allow for comparison and repeatability of the experiment, the results we reported in the main part of this work were averaged over 100 runs of the experiment. The dataset generated for each run consists of positive triples, extracted from PhosphoSitePlus, and negative triples, generated by randomly combining kinases with (site,substrate) pairs that appear in PhosphoSitePlus. The training split accounts for 90% of the data, the remaining 10% is used for testing. Both training and test set contain equal numbers of positive and negative instances.

To evaluate LinkPhinder, training data are used to learn a model in each run and its performance is evaluated on the test set. Triples in the test set are assigned the prediction score if this is available, otherwise a zero score is assigned.

One note to be taken into account regarding prediction score assignment is this. As stated in the main text, a very accurate model that generates predictions only on a small subset of the triples may be of limited use in phosphorylation prediction. Hence, we also assessed the rate of predictions a model is able to generate by measuring the percentage of triples in the test set for which the model is able to generate a prediction (i.e. non-zero score). We refered to this value as *coverage* in the Results section.

Concerning the existing methods to which we compare ourselves, scores are extracted from the predictions provided by each method (created as described in the previous section). This does not exclude that part of the testing triples may have been used to train the comparative models. Assuming that this is the case, this would represent a disadvantage in terms of performance for our model. Similarly to the LinkPhinder case, coverage is therefore computed over the test data and zero scores are assigned to triples for which a prediction is not available.

**Verifying the stability of LinkPhinder under different conditions of the computational experiments.** To make sure various decisions made in preparation of the benchmarking data do not influence the presented results in terms of comparing the performance of LinkPhinder and related existing tools, we have first experimented with a different positive-negative ratio (ten negatives per one positives, see Table 9), and then with various different train-test split ratios (Table 10).

The increase in the number of negatives per a positive typically hampers performance of ranking-based models, and Table 9 clearly shows that our experiments are no exception. However, one can also immeditaly notice that LinkPhinder remains by far the best tool, and is significantly less affected by the change. This demonstrates the superior stability of our tool in the context of changing experimental conditions.

The results in Table 10 clearly show that while the performance of LinkPhinder decreases with increasing proportion of testing over the training data, it is still superior to the corresponding

**Table 9. LinkPhinder performance compared to other systems on our benchmark with 1:10 positive to negative ratio in the testing split where the training/testing splits are 90% and 10% respecitvely.**

| Model | AUPR | AUROC | P@10 | P@50 |
|---|---|---|---|---|
| GPS | 0.259 ± 0.007 | 0.731 ± 0.006 | 0.337 ± 0.145 | 0.416 ± 0.063 |
| NetworKin | 0.281 ± 0.009 | 0.618 ± 0.007 | 0.798 ± 0.122 | 0.756 ± 0.055 |
| NetPhorest | 0.199 ± 0.007 | 0.597 ± 0.007 | 0.542 ± 0.137 | 0.520 ± 0.071 |
| Scansite | 0.149 ± 0.004 | 0.571 ± 0.006 | 0.132 ± 0.099 | 0.210 ± 0.048 |
| Phosphopredict | 0.091 ± 0.002 | 0.500 ± 0.006 | 0.029 ± 0.050 | 0.050 ± 0.029 |
| Netphos | 0.166 ± 0.006 | 0.563 ± 0.007 | 0.426 ± 0.149 | 0.390 ± 0.064 |
| LinkPhinder | **0.875 ± 0.010** | **0.982 ± 0.002** | **0.993 ± 0.025** | **0.981 ± 0.024** |

**Table 10. Relative LinkPhinder performance across different training-testing splits where the positive to negative ratio of the testing set is 1:10 (the relative performance results were substantially less variable for the 1:1 ratio, therefore we do not report them here).**

| Model | AUPR | AUROC | P@10 | P@50 |
|---|---|---|---|---|
| Train 60%, Test 40% | 0.768 ± 0.006 | 0.969 ± 0.001 | 0.987 ± 0.034 | 0.981 ± 0.017 |
| Train 70%, Test 30% | 0.797 ± 0.006 | 0.974 ± 0.001 | 0.960 ± 0.049 | 0.968 ± 0.018 |
| Train 80%, Test 20% | 0.835 ± 0.005 | 0.978 ± 0.001 | 0.990 ± 0.030 | 0.984 ± 0.012 |
| Train 90%, Test 10% | 0.875 ± 0.010 | 0.982 ± 0.002 | 0.993 ± 0.025 | 0.981 ± 0.024 |

results of the related works given in Table 9. This further corroborates our claim of LinkPhinder's stability with respect to different experimental conditions.

**Generating phosphorylation data for the web interface of LinkPhinder.** In order to prepare data of phosphorylation reactions for prediction, a list of known kinases and a list of known substrates with their corresponding phosphorylation sites are extracted from the phosphorylation network. The elements of the lists are combined using their Cartesian product to generate every possible combination of kinase and phosphorylation site of each substrate. These are converted into knowledge graph phosphorylation statements and are then scored using the previously trained best-performing prediction model (*i.e.* the result of the grid search described previously). Finally, knowledge graph statements with their associated scores are converted back to phosphorylation site-specific statements. If there are duplicate statements after the conversion process that only differ in the scores assigned to them by the conversion and the model, we only keep the one with the highest score determined by the model. This is motivated by the fact that the model utilises more information on the actual phosphorylations than the conversion process and therefore its scores override the scores assigned after conversion.

## Experimental model and subject details

**Cell culture experiments for targeted validation.** Hek-293 cells were regularly grown in Dulbecco's modified medium supplemented with 10% foetal serum. Subconfluent cell were transfected with Lipofectamine (Invitrogene) following manufacturer's instructions. pSG5-gag-AKT was previously described [37]. LATS1 siRNA and AKT siRNA were from Dharmacorn and sequences have been described before [29]. Twentyfour hours after tranfection HEK293 cells were serum deprived for 16 hours. Subsiquently, cell were lysed in 20mM HEPES (pH 7.5), 150 mM NaCl, 1% NP-40, phosphatase inhibitors (2mM NaF, 10mMb-Glycerolphosphate, 2 MM Na4P2O4) and protease inhibitors (5 $\mu$g/ml Leupeptin and 2.2 $\mu$g/ml aprotinin). Cell lysates were separated by SDS-PAGE analysed by western blotting. Phosphorylated proteins were immunoprecipitated with pAKT-Substrate specific antibody. Briefly, the lysates were incubated with 1$\mu$l of antibody and 5$\mu$l of protein-G sepharose beads for 1 hour at 4C in an orbital wheel. The immunoprecipitates were washed 3 times with lysis buffer. 2 bed volumes of denaturing laemli buffer were added to the dry pelleted beads and immunocomplex were eluted by boiling the samples at 100C for 5 minutes. Anti-creb, anti-LATS1 anti-P53 anti-tub, p-YAP-S127 were obtained from commercial sources.

**Mass-spectrometry experiments for extended validation.** HeLa cells were transiently transfected with a GFP-tagged LATS1 construct or a GFP construct as control. After 2 days they were serum starved over-night and left untreated (control) or were treated with FasL (50nM) or Etoposide (50$mu$M) for 16 hours. Then, cells were lysed with Lysis buffer (20mM 4-(2 hydroxyethyl)-1piperazineethanesulfonic acid (HEPES) pH7.5, 150mM NaCl, 1% NP-40, phosphatase inhibitors (10 mM$\beta$-Gycerolphosphate, 1 mM $Na_3VO_4$, 2mM $Na_4P_2O_7$, 2 mM NaF) and protease inhibitors (5 $\mu$g/ml Leupeptin and 2.2 $\mu$g/ml Aprotinin), and proteins were immunoprecipitated using GFP-trap_A (Chromotek) according to the manufacturer's instructions. The beads were washed 3 times with lysis buffer followed by two washes with the same buffer not containing NP-40. The proteins immunoprecipitated onto GFP-beads were prepared for masss-spectrometry analysis as previously described [38]. Briefly, the immunoprecipitates were digested in two steps. Firstly, by adding 60$\mu$l of elution buffer-1 (2M urea, 50mM Tris-HCl pH7.5, 5$\mu$g/ml Trypsin), to each sample and incubation at 27˚C on a shaker. After 30 minutes initial digestion the samples were centrifuged at 13,000 rpm in a table top centrifuge for 30 seconds and the supernatant was collected into a new Eppendorf tube. In the

second step 25$\mu$l of elution buffer-2 (2M urea, 50mM Tris-HCL pH7.5, 1mM Dithiothreitol) was added per sample followed by centrifugation as above. The supernatant was collected into a new Eppendorf tube. The elution step was repeated, and both supernatants were combined and incubated overnight at room temperature to allow trypsin digestion to go to completion. The, samples were alkylated by adding 20 $\mu$l iodoacetamide (5mg/ml), and incubation for 30 min in the dark at room temperature. The reaction was stopped by adding 1 $\mu$l 100% Trifluoracetic acid (TFA) to each sample. 100 $\mu$l of each sample was immediately loaded into equilibrated handmade C18 StageTips containing Octadecyl C18 disks (Supelco) for desalting. Tips were previously activated by washing with 50$\mu$l of 50% AcN and 0.1%TFA. After a quick centrifugation the tips were washed with 50$\mu$l of 0.1%TFA. 100$\mu$l samples was loaded onto the tip washed twice with 50$\mu$l of 0.1% TFA and eluted twice with 25$\mu$l of 50% AcN and 0.1% TFA solution. The eluates were combined and concentrated until the volume was reduced to 5$\mu$l using a CentriVap Concentrator (Labconco). Samples were diluted to obtain a final volume of 15$\mu$l by adding 0.1% TFA and centrifuged for 10 minutes at 13000rpm. 12$\mu$l of the samples were analysed by MS. The samples were analysed by liquid Chromatography-Tandem Mass Spectrometry (Nanoflow Ultimate 3000 LC and Q-Exactive mass spectrometer [Thermo]). A 10 cm long, 75 $\mu$m inner diameter, HLPC c18-reversed pahes column was used. Samples were loaded at 600nl/min and peptides were eluted at a constant flow rate of 250nl/ min for 40 min. A multisegment linear gradient of 2-135% buffer (98% Acetonitrile and 0.1% formic acid) in positive ion mode was used. Data were acquired with the mass spectrometer operating in automatic data dependent switching mode selecting the 12 most intense ions prior to MS/MS analysis. Mass spectra were analysed by MaxQuant. Label-free quantitation was performed using MaxQuant.

**PKA Kinase assay**. Serum straved HEK293T were lysed in a Nonidet P-40 buffer (50 mM Tris-HCl, pH 7.8, 150 mM NaCl, 1% (vol/vol) Nonidet P-40, protease inhibitors and phosphatase inhibitors). Lysates were treated at 1 mg/ml with 10 mM 5'-4-fluorosulphonylbenzoyladenosine (FSBA) solubilised in DMSO and then incubated at 31˚C for 2 hour. Samples were spun down at 200 x g to remove any precipitate. Sample were diluted down with 2 ml of PKA kinase buffer (50 mM Tris pH 7.5, 10 mM MgCl2, 0.1 mM EGTA and 2 mM DTT) and desalted using a Millipore Amicon ultrafiltration columns with a 3 kDa molecular weight cutoff. Following concentration, the samples were incubated with PKA kinase buffer (50 mM Tris pH 7.5, 10 mM MgCl2, 0.1 mM EGTA and 2 mM DTT), 500 uM ATP-biotin and 1250 units of recombinant PKA (New England Biolabs) in a total volume of 60 $\mu$l. Control samples without recombinant PKA and ATP-biotin were also made up. The controls and kinase-added samples were incubated at 31˚C for 2 hours. 300 $\mu$l of phosphate buffer was added to the samples. Streptavidin resin (100 $\mu$l of a 50% slurry) was incubated with the samples overnight at 4˚C. Samples were spun down samples at 2000 x g for 1 minute and the supernatant was removed. Samples were washed 5 times with 1 ml of phosphate buffer. Samples were analysed by mass spectrometry.

The full results of the assay are given in the kinase assays supplement (S1 Table).

**MST2 Kinase assay**. Serum straved HEK293T cells were treated with 3 $\mu$M of the MST2 kinase specific inhibitor, XMU-MP-1 or DMSO for 3 hours. Cells were lysed in a Nonidet P-40 buffer (50 mM Tris-HCl, pH 7.8, 150 mM NaCl, 1% (vol/vol) Nonidet P-40, protease inhibitors and phosphatase inhibitors). Lysates were treated at 1 mg/ml with 10 mM 5'-4-fluorosulphonylbenzoyladenosine (FSBA) solubilised in DMSO and then incubated at 31˚C for 2 hour. Samples were spun down at 200 x g to remove any precipitate. Sample were diluted down with 2 ml of MST2 kinase buffer (40 mM HEPES pH 8.0, 10 mM MgCl2, 0.5 mM EGTA) and desalted using a Millipore Amicon ultrafiltration columns with a 3 kDa molecular weight cutoff. Following concentration, the samples were incubated with MST2 kinase assay buffer (40

mM HEPES pH 8.0, 10 mM MgCl2, 0.5 mM EGTA), 500 uM ATP-biotin and 32 ng of recombinant MST2 (made in house) in a total volume of 60 $\mu$l. Control samples without recombinant MST2 and ATP-biotin were also made up. The controls and kinase-added samples were incubated at room temperature for 3 hours. 300 $\mu$l of phosphate buffer was added to the samples. Streptavidin resin (100 $\mu$l of a 50% slurry) was incubated with the samples for 1 hour at room temperature. Samples were spun down samples at 2000 x g for 1 minute and the supernatant was removed. Samples were washed 5 times with 1 ml of phosphate buffer. Samples were analysed by mass spectrometry.

The full results of the assay are given in the kinase assays supplement (S2 Table).

**LATS1 Kinase assay**. Serum straved HEK293T cells were lysed in a Nonidet P-40 buffer (50 mM Tris-HCl, pH 7.8, 150 mM NaCl, 1% (vol/vol) Nonidet P-40, protease inhibitors and phosphatase inhibitors). Lysates were treated at 1 mg/ml with 10 mM 5'-4-fluorosulphonyl-benzoyladenosine (FSBA) solubilised in DMSO and then incubated at 31˚C for 2 hour. Samples were spun down at 200 x g to remove any precipitate. Sample were diluted down with 2 ml of LATS1 kinase buffer (25 mM HEPES pH 7.4, 50 mM NaCl, 5 mM MgCl2 and 5 mM MnCl2, 5 mM $\beta$-glycerophosphate and 1 mM dithiothreitol) and desalted using a Millipore Amicon ultrafiltration columns with a 3 kDa molecular weight cutoff. Following concentration, the samples were incubated with LATS1 kinase assay buffer (25 mM HEPES pH 7.4, 50 mM NaCl, 5 mM MgCl2 and 5 mM MnCl2, 5 mM $\beta$-glycerophosphate and 1 mM dithiothreitol), 500 uM ATP-biotin and 100 ng of recombinant LATS1 (Abcam) in a total volume of 60 $\mu$l. Control samples without recombinant LATS1 and ATP-biotin were also made up. The controls and kinase-added samples were incubated at 30˚C for 30 minutes. 300 $\mu$l of phosphate buffer was added to the samples. Streptavidin resin (100 $\mu$l of a 50% slurry) was incubated with the samples for 1 hour at room temperature. Samples were spun down samples at 2000 x g for 1 minute and the supernatant was removed. Samples were washed 5 times with 1 ml of phosphate buffer. Samples were analysed by mass spectrometry.

The full results of the assay are given in the kinase assays supplement (S3 Table).

**Mass spectrometry sample preparation**. The streptavidin resin containing the bound proteins were incubated with 400 $\mu$l of elution buffer I (50 mM Tris-HCl ph 7.5, 2 M Urea, 181 ng/$\mu$l trypsin) at 37˚C for 30 minutes. The samples were spun at 2000 x g and the superntant was retained. To the streptavidin resin 330 $\mu$l of elution buffer II (50 mM Tris-HCl ph 7.5, 2 M Urea, 1 mM DTT) at 37˚C for 1 hour. The samples were spun at 2000 x g and the superntant was retained. The two supernatant of elution buffers I and II were combined and incubated overnight at 37˚C. After the incubation 130 $\mu$l of 5 mg/ml Iodocetamide was added to each and the samples were incubated for 30 minutes at room temperature in the dark. C18 stage tips that were previously prepared were mounted into a 1.5 ml eppendorf were activated by adding 50 $\mu$l of 50% acetonitrile (AcN) and 0.1% Trifluoroacetic acid (TFA). The samples were spun at 5000 rpm for 1 minute. 50 $\mu$l of 1% TFA was added to the C18 stage tips and the samples were spun at 5000 rpm. After the Iodocetamide incubation the reaction was stopped by adding 1 $\mu$l of 100% TFA. The samples were loaded onto the C18 stage tips and they were spun at 5000 rpm. The C18 stage tips were then washed by adding 50 $\mu$l of 1% TFA and then the samples were spun at 5000 rpm, this was done twice. Before elution of the samples, the C18 stage tips were mounted into fresh 1.5 ml eppendorfs. The peptides were eluted of the C18 stage tips by adding 25 $\mu$l of 50% AcN and 0.1% TFA and spinning the samples at 5000 rpm, this was repeated twice. Samples were evaporated for 10-15 in a CentriVap concentrator until 5 $\mu$l was left. The sample was then respuspended in 20 $\mu$l of TFA. The samples were then analysed by mass spectrometry.

**Mass spectrometry**. Mass spectrometry was performed using a Ultimate 3000 RSLC system that was coupled to an Orbitrap Fusion Tribrid mass spectrometer (Thermo Fisher Scientific).

Following tryptic digest, the peptides were loaded onto a nano-trap column (300 $\mu$M i.d x 5mm precolumn that was packed with Acclaim PepMap100 C18, 5 $\mu$M, 100 Å; Thermo Scientific) running at a flow rate of 30 $\mu$l/min in 0.1% trifluoroacetic acid made up in HPLC water. The peptides were eluted and separated on the analytical column (75 $\mu$M i.d. × 25 cm, Acclaim PepMap RSLC C18, 2 $\mu$M, 100 Å; Thermo Scientific) after 3 minutes by a linear gradient from 2% to 30%of buffer B (80% acetonitrile and 0.08% formic acid in HPLC water) in buffer A (2% acentonitrile and 0.1% formic acid in HPLC water) using a flow rate if 300 nl/min over 150 minutes. The remaining peptides were eluted using a short gradient from 30% to 95% in buffer B for 10 minutes. The mass spectrometry parameters were as follows: for full mass spectrometry spectra, the scan range was 335-1500 with a resolution of 120,000 at m/z = 200. MS/MS acquisition was performed using top speed mode with 3 seconds cycle time. Maximum injection time was 50 ms. The AGC target was set to 400,000, and the isolation window was 1 m/z. Positive Ions with charge states 2-7 were sequentially fragmented by higher energy collisional dissociation. The dynamic exclusion duration was set at 60 seconds and the lock mass option was activated and set to a background signal with a mass of 445.12002.

**Analysis of mass spectrometry data**. Analysis was performed using MaxQuant (version 1.5.3.30). Trypsin was set to be the digesting enzyme with maximal 2 missed cleavages. Cysteine carbmidomethylation was set for fixed modifications and oxidation of methionine and N-thermal acetylation were specified as variable modifications. The data was then analysed with the minimum ratio count of 2. The first search peptide was set to 20, the main search peptide tolerance to 5 ppm and the "re-quantify" option was selected. For protein and peptide identification the Human subset of the SwissProt database (Release 2015_12) was used and the contaminants were detected using the MaxQuant contaminant search. A minimum peptide number of 1 and a minimum of 6 amino acids was tolerated. Unique and razor peptides were used for quantification. The match between run option was enabled with a match time window of 0.7 minutes and an alignment window of 20 minutes.

## Quantification and statistical analysis

**Peptide identification.** MaxQuant (version 1.3.0.5.) was used to analyse raw mass spectrometric data files from LC-MS/MS for protein quantification. Default settings were used unless stated otherwise, including the following parameters: Trypsin/P digest allowing for 2 misscleavages; variable modifications included oxidation and acetylation; fixed modification included carbamidomethylation (at Cysteine); to detect phosphopeptides we included phospho (STY) as a modification; first search at 20 ppm: main search at 6 ppm mass accuracy (MS) and 20ppm mass deviation for the fragment ions. The MS data were searched against a human database (Uniprot HUMAN) with a minimum peptide length of 6, unfiltered for labelled amino acids, at a false discovery rate (FDR) of 0.01 for peptides and proteins. The results were refined through the re-quantify option; also "match between runs" was selected with a 1 min time window, and label free quantification was selected with the minimum ratio count set at 1.

## Supporting information

**S1 Table. PKA Kinase Assay Results (an PDF file; c.f. https://doi.org/10.6084/m9.figshare. 13118441).**
(PDF)

**S2 Table. MST2 Kinase Assay Results (an PDF file; c.f. https://doi.org/10.6084/m9. figshare.13118477).**
(PDF)

**S3 Table. LATS1 Kinase Assay Results (an PDF file; c.f. https://doi.org/10.6084/m9. figshare.13118483).**
(PDF)

**S4 Table. Mass-spec results for the LATS1 IP (an xlsx file; c.f. https://doi.org/10.6084/m9. figshare.12173163).**
(XLSX)

**S5 Table. Mass-spec data for the PKA kinase assay (an xlsx file; c.f. https://figshare.com/ articles/Mass_spec_data_PKA/12200681).**
(XLSX)

**S6 Table. Mass-spec data for the MST2 kinase assay (an xlsx file; c.f. https://doi.org/10. 6084/m9.figshare.12200675.v1).**
(XLSX)

**S7 Table. Mass-spec data for the LATS1 kinase assay (an xlsx file; c.f. https://figshare.com/ articles/Mass_spec_data_LATS_kinase_assay/12200597).**
(XLSX)

**S1 Data. Full set of LinkPhinder predictions (a single bzip2-archived CSV file; c.f. https:// doi.org/10.6084/m9.figshare.12173100).**
(BZ2)

**S2 Data. Full set of predictions computed by the related works (a bzip2-archive of 6 CSV files for each of the related tools; c.f. https://doi.org/10.6084/m9.figshare.12173109).**
(TBZ)

**S1 Fig.** Supporting details on the experimental validation of the LATS1/YAP1 phosphorylation: (A-B) HEK293 were transfected with the indicated siRNAs. 48 hours after transfection the cells were lysed and blotted with the indicated antibodies. (C) HEK293 were transfected with empty vector (EV) or GAG-AKT or treated with AKTi IV (10M) for 1 hour. Phosphorylated proteins were immunoprecipitated using an anti-AKT antibody and the immunoprecipitates were blotted with the indicated antibodies (a PDF figure, c.f. https://doi.org/10.6084/m9. figshare.13118561).
(PDF)

## Author Contributions

**Conceptualization:** Vít Nováček, Pierre-Yves Vandenbussche, Walter Kolch, Dirk Fey.

**Data curation:** Piero Conca, Emir Muñoz, Luca Costabello, Kamalesh Kanakaraj, Zeeshan Nawaz, Pierre-Yves Vandenbussche.

**Funding acquisition:** Vít Nováček, David Matallanas, Pierre-Yves Vandenbussche, Walter Kolch, Dirk Fey.

**Methodology:** Vít Nováček, David Matallanas, Pierre-Yves Vandenbussche, Walter Kolch, Dirk Fey.

**Project administration:** Vít Nováček, Pierre-Yves Vandenbussche, Walter Kolch.

**Resources:** Gavin McGauran, David Matallanas, Adrián Vallejo Blanco, Walter Kolch, Dirk Fey.

**Software:** Vít Nováček, Piero Conca, Emir Muñoz, Luca Costabello, Kamalesh Kanakaraj, Zeeshan Nawaz, Sameh K. Mohamed.

**Supervision:** Vít Nováček, Pierre-Yves Vandenbussche, Walter Kolch.

**Validation:** Vít Nováček, Gavin McGauran, David Matallanas, Adrián Vallejo Blanco, Piero Conca, Emir Muñoz, Luca Costabello, Kamalesh Kanakaraj, Zeeshan Nawaz, Brian Walsh, Sameh K. Mohamed, Pierre-Yves Vandenbussche, Dirk Fey.

**Visualization:** Kamalesh Kanakaraj, Zeeshan Nawaz.

**Writing – original draft:** Vít Nováček, Pierre-Yves Vandenbussche, Dirk Fey.

**Writing – review & editing:** Colm J. Ryan, Walter Kolch.

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
