## [Decision Letter · Decision Letter 0]

27 Feb 2020

Dear Dr Novacek,

Thank you very much for submitting your manuscript "Accurate Prediction of Kinase-Substrate Networks Using Knowledge Graphs" for consideration at PLOS Computational Biology.

As with all papers reviewed by the journal, your manuscript was reviewed by members of the editorial board and by several independent reviewers. In light of the reviews (below this email), we would like to invite the resubmission of a significantly-revised version that takes into account the reviewers' comments.

We cannot make any decision about publication until we have seen the revised manuscript and your response to the reviewers' comments. Your revised manuscript is also likely to be sent to reviewers for further evaluation.

Sincerely,

Anand R. Asthagiri

Associate Editor

PLOS Computational Biology

Douglas Lauffenburger

Deputy Editor

PLOS Computational Biology

Reviewer's Responses to Questions

**Comments to the Authors:**

Reviewer #1: The authors present a novel approach based on knowledge graphs to predict kinase-substrate interactions that benchmarks favourably against well established methods in the field. Moreover, they experimentally confirm several of the top predicted interactions. This work tackles an important challenge and utilises a methodological distinct method that expands substrate predictions to many kinases that are not covered by existing approaches.

While this approach represents a significant development the method needs to be benchmarked using independent data-sets to better understand the precision of its predictions. Moreover, in several sections conclusions are not well supported. Thus, it would be important to expand the current set of figure panels and supplementary materials to provide clear evidence supporting the conclusions made.

1. It would be important to exclude that the precisions reported are not inflated due to overfitting to the training data-set. Besides the 90-10% train-test split, can the authors use, for example, 60-40% and 80-20%?

2. Taking advantage of a recent systematic study of kinase-substrate interactions (PMID:31959955), the authors could use this as an independent test set, while making sure the version of PhosphositePlus used for training does not contain this study to avoid circular training-testing. Can the authors show how well LinkPhinder captures these interactions and how it compares to the other approaches?

3. LinkPhinder identifies 2,009,171 and 7,232,636 high and medium stringency interactions, respectively. It would be important to see overall statistics of the predicted kinase-substrate networks of each method, for example, the number of kinases covered, distribution and average number of predicted substrates per kinase, distribution and average number of kinases targeting each phosphorylation-site.

4. Some sections of the manuscript, for example section 2.3 and 2.4, have statements that are not supported with links to either figures, tables or supplementary material. I would recommend the authors to expand the number of figure panels accordingly to provide better support to the text. Also, data not shown statements are problematic to assess and reproduce. I would suggest either to provide the data as supplementary information or remove this statement.

5. Improved kinase-substrate networks are fundamental for methods that aim to estimate kinase activity profiles based on their substrates phosphorylation status. It would be interesting to see if LinkPhinder predicted network improves kinase activity predictions. An existing benchmark data-set (PMID:28200105) of quantitative phosphoproteomics measurements upon kinase inhibition could be used to test this.

Minor revisions

1. Is the predictive power of LinkPhinder networks significantly different between each run? How would LinkPhinder benchmark against other methods if instead of 100 runs of the experiment only one was performed?

2. Table 1 could be complemented with plots of the AU-PR and AU-ROC as supplementary figures.

3. Typo in “equal” in Table 1 legend.

**Have all data underlying the figures and results presented in the manuscript been provided?**

Reviewer #1: Yes

PLOS authors have the option to publish the peer review history of their article (what does this mean?). If published, this will include your full peer review and any attached files.

Reviewer #1: No
---

## [Decision Letter · Decision Letter 1]

27 Jun 2020

Dear Dr Novacek,

Thank you very much for submitting your manuscript "Accurate Prediction of Kinase-Substrate Networks Using Knowledge Graphs" for consideration at PLOS Computational Biology. As with all papers reviewed by the journal, your manuscript was reviewed by members of the editorial board and by several independent reviewers. The reviewers appreciated the attention to an important topic. Based on the reviews, we are likely to accept this manuscript for publication, providing that you modify the manuscript according to the review recommendations.

Sincerely,

Anand R. Asthagiri

Associate Editor

PLOS Computational Biology

Douglas Lauffenburger

Deputy Editor

PLOS Computational Biology

[LINK]

Reviewer's Responses to Questions

**Comments to the Authors:**

Reviewer #1: The authors have substantially addressed my comments and I believe this method is of overall interest for the community, although I still have some significant concerns. I found it hard to navigate through the changes because point-by-point replies are mostly pointers to the manuscript and the generated PDF seems to have the manuscript included three times, with revised and unrevised versions, hence the total pages of 258.

Despite the fact that LinkPhinder outperforms all other methods, the poor results with the independent data-set, with AUPR and AUROC < 0.54 for all methods (Table 3), raises some substantial concerns regarding the performance of LinkPhinder in the internal validation (Table 1). It is important to understand if this is (i) an inherent problem of the validation data-set; (ii) potential overfitting of LinkPhinder to the test data-set; or (iii) methodological problem of the validation, e.g. data leakage from train to test.

Can the authors comment on the fact that PhosphoSitePlus is both used to train LinkPhinder and to generate the true positive sets for the AUPR and AUROC curves. This could lead to data leakage between train and test, i.e. training data is also used to test. Could the authors generate random sets of positive/negative interactions that are kept from LinkPhinder training and use them only for validation and comparison with other methods? In addition, please report the total number of positive and negative statements (ideally have the list as supplementary material), add details to the methods on how these are generated, and plot the AUPR and AUROC curves with the mean curve obtained for each model.

**Have all data underlying the figures and results presented in the manuscript been provided?**

Reviewer #1: Yes

PLOS authors have the option to publish the peer review history of their article (what does this mean?). If published, this will include your full peer review and any attached files.

Reviewer #1: No
---

## [Editor Report · Decision Letter 2]

10 Aug 2020

Dear Dr Novacek,

We are pleased to inform you that your manuscript 'Accurate Prediction of Kinase-Substrate Networks Using Knowledge Graphs' has been provisionally accepted for publication in PLOS Computational Biology.

Best regards,

Anand R. Asthagiri

Associate Editor

PLOS Computational Biology

Douglas Lauffenburger

Deputy Editor

PLOS Computational Biology

---

## [Editor Report · Acceptance letter]

23 Nov 2020

PCOMPBIOL-D-19-02065R2 

Accurate Prediction of Kinase-Substrate Networks Using Knowledge Graphs

Dear Dr Novacek,

I am pleased to inform you that your manuscript has been formally accepted for publication in PLOS Computational Biology. Your manuscript is now with our production department and you will be notified of the publication date in due course.

With kind regards,

Matt Lyles
